



# The Vertical Variability of Black Carbon Observed in the Atmospheric Boundary Layer during DACCIWA

Barbara Altstädter[1], Konrad Deetz[2], Bernhard Vogel[2], Karmen Babić[2], Cheikh Dione[2],
Federica Pacifico[3], Corinne Jambert[3], Friederike Ebus[1], Konrad Bärfuss[1], Falk Pätzold[1],
Astrid Lampert[2], Bianca Adler[2], Norbert Kalthoff[2], and Fabienne Lohou[3]

[1]Institute of Flight Guidance, Technische Universität Braunschweig, Braunschweig, Germany
[2]Institute of Meteorology and Climate Research, Karlsruhe Institute of Technology, Karlsruhe, Germany
[3]Laboratoire d'Aérologie, Université de Toulouse, CNRS, UPS, Toulouse, France

**Correspondence:** Barbara Altstädter (b.altstaedter@tu-braunschweig.de)

**Abstract.** The vertical variability of the black carbon (BC) mass concentration in the atmospheric boundary layer (ABL) is analysed during the West-African Monsoon (WAM) season. BC was measured with a micro aethalometer integrated in the payload bay of the unmanned research aircraft ALADINA (Application of Light-weight Aircraft for Detecting IN situ Aerosol) as part of the field experiment of the DACCIWA (Dynamics-Aerosol-Chemistry-Cloud Interactions in West Africa) project.

In total, 53 measurement flights were performed at the local airfield of Savè, Benin, in the period of 2–16 July 2016. The mean results show a high variability of BC (1.79 to 2.42 $\pm 0.31\,\mu\mathrm{g\,m^{-3}}$) influenced by the stratification of the ABL during the WAM. The model COSMO-ART (Consortium for Small-scale Modelling–Aerosols and Reactive Trace gases) was applied for the field campaign period and used in order to investigate possible sources of the measured BC. The model output was compared with the BC data on two selected measurement days (14 and 15 July 2016). The modeled vertical profiles of BC

show that the observed BC was already altered, as the size was mainly dominated by the accumulation mode. Further, the calculated vertical transects of wind speed and BC showed that the measured BC layer was transported from the south with maritime inflow, but was mixed vertically after to the onset of the nocturnal low-level jet (NLLJ) at the measurement site. The validations and the ground observations of gas concentrations ($NO_x$ and CO) confirm that primary emission could be excluded during the case study, in contrast to initially expected. The case underlines the important role of BC transport processes in the

WAM area.

## 1 Introduction

BC is one of the major contributors affecting the Earth's climate system. Freshly emitted BC caused by the incomplete combustion, is insoluble in water and it strongly absorbs the solar radiation in the visible spectrum (Bond et al., 2013). However, if BC is once emitted into the atmosphere, ongoing physical and chemical reactions, like the secondary aerosol formation influence

the composition and the mixing state of the aerosol particles. BC might develop to a possible actuator as cloud condensation nuclei (CCN), as hygroscopic growth enables coating with soluble species in the atmosphere (Zhang et al., 2008). The understanding of the BC life cycle is still poor, as the monitoring of the type and source of combusted products, the ageing processes





in the atmosphere and the wet deposition differ in the used measurement methods, and simulations show a high discrepancy (e.g., Li et al., 2017). The role of BC emission is of high interest in regions with dense population. For instance, the population in southern West Africa (SWA) is proposed to grow to more than 800 million by the middle of the century (Knippertz et al., 2015a), consequently a degradation of the air quality and the human health is expected by ongoing anthropogenic pollutions.

Liousse et al. (2014) compared three different future scenarios for the African emissions in the year 2030, if no regulations for the combustions will be carried out. The BC emission factors varied between 2 and 4 in comparison with 2005, meaning a possible emission of more than 2.5 Tg BC y$^{-1}$ that would be comparable with 20–30 % of the BC emissions worldwide.

The impact of the growing air pollution on the SWA area at a regional and global scale is one subject of the DACCIWA project (see Knippertz et al., 2015b). The project combines in-situ and aircraft observations that are further compared with

satellite data and models in order to achieve a better understanding of weather, clouds and climate of SWA that is suffering from a weak monitoring network over a large area in comparison with other continental sites, e.g. in Europe and North America. The project aims at understanding complex interactions between aerosols, clouds and their impact on the climate and weather in SWA. The intensive field experiment of DACCIWA was conducted in June–July 2016 (e.g., Knippertz et al., 2017; Flamant et al., 2018; Kalthoff et al., 2018). During the study, the investigation area was affected by the occurrence of nocturnal low-level

jets (NLLJ) and low-level clouds (LLC) as consequence of the WAM season (e.g., Dione et al., 2018; Kalthoff et al., 2018; Adler et al., 2019; Babić et al., 2019a). The occurrence of LLC during the monsoon are of special interest, as they affect the diurnal cycle of the ABL (Gounou et al., 2012) and their formation processes have to be integrated into models as well, as stated by Adler et al. (2017) and Hannak et al. (2017).

The aerosol composition in the WAM region is mainly influenced by the large-scale synoptic pattern (Mari et al., 2008).

During July and September, biomass burning smokes from southern Africa are the dominant contributors during WAM season and were proposed as a major source for BC, retrieved by satellite data and simulations (Huang et al., 2009). Horizontal transects were performed with aircraft at the coastal site of SWA and the results of the PM$_1$ concentration were shown in Brito et al. (2018). Interestingly, the observed BC fraction of the aerosol composition varied only minimal (6 %) above in-plume flights in comparison with the background concentration of BC with 0.34 µg m$^{-3}$ in the lowermost 2 km. The observations are

in agreement with the simulation results by Huang et al. (2009), showing that BC concentration is more likely dominated by long range transport triggered by the WAM instead of local biomass burning emissions.

However, previous data showed only the large scale variability of BC instead of local effects on smaller scales of a few square kilometres at one measurement field. Therefore, a large benefit can be expected by using unmanned aerial systems (UAS) for investigating the small scale variability of BC. There is a wide range of airborne platforms, including UAS operating with

batteries and fuel engines, both fixed wings and multicopters that are used for observing the atmosphere with high resolution on small scales (e.g., Mayer et al., 2010; Marino et al., 2015; Renard et al., 2016; Jiménez et al., 2016; Cuxart et al., 2016; Båserud et al., 2016; Brosy et al., 2017). In order to quantify the small-scale vertical BC distribution and to understand the role of small-scale local emissions, the UAS ALADINA of aircraft type Carolo P360 (Altstädter et al., 2015) was operated during the field experiment of DACCIWA in Savè, Benin, between 2 and 16 July 2016. The ALADINA measurement period was in

the so called Phase 2 lasting from the post-onset until the vortex phase of the WAM season, as stated by Knippertz et al. (2017).



The first part of the period was influenced by LLC with drier periods during the vortex phase that was predominant from 9 until 17 July 2016 (Kalthoff et al., 2018). ALADINA measurements enabled the characterisation of BC mass concentrations in relation to meteorological parameters of the ABL.

The objective of the study is to investigate the small scale vertical variability of BC at a remote site in West Africa, in general for the whole observation period, and in particular for a case study during the vortex phase with enhanced BC concentration and reduced humidity induced by long-range transport, which are unusual conditions during the monsoon period. Besides, the aim is to show the capability of an aethalometer newly implemented into the UAS. To the authors' knowledge, there is no study of the vertical distribution of BC obtained by unmanned research aircraft in the WAM region so far. In addition to BC

measurements, the COSMO-ART model is used in order to obtain information about the particle size and distribution. Further, the ground observations of wind profiler, ceilometer and gas concentrations are taken into account that were installed 4 km away from the research flights in order to characterise the current weather situation and possible sources for BC.

The article is structured as follows: The UAS and the instrumentation integrated in the payload bay of ALADINA are introduced in Sect. 2. The description of the AE51 and the applied methods during post-processing are given in Subsect.

2.3. The model set up of COSMO-ART is provided in Subsect. 2.4 and the measurement site of the research flights and the additional ground observations at the Savè supersite are presented in Subsect. 2.5. The results are discussed in Sect. 3 and the outcome of the study is summarised in Sect. 4.

## 2    Instrumentation, Model, and Measurement Site

### 2.1    The Unmanned Aerial System ALADINA

The unmanned research aircraft ALADINA was designed at the Technische Universität Braunschweig for atmospheric research and was steadily improved regarding the current technical requirements (Fig. 1). The Carolo P360 has a maximum take off weight of 25 kg and a wing span of 3.6 m. The modular payload bay offers the capacity for a payload weight of approximately 4 kg. The pusher aircraft has an electrical propulsion and enables a flight endurance of 40–50 min at a cruising speed of 28–30 m s$^{-1}$ in the current configuration. ALADINA is operated manually during take off and landing, whereas controlled

automatically during the measurement flights via autopilot. With the installed GPS (Global Positioning System) and IMU (Inertial Measurement Unit) the flight time, the accurate position, altitude and the attitude of the aircraft are given. Besides, a precision of $\pm 1.5$ m in constant height is provided by the autopilot system. The flight track follows predefined waypoints that are sent before take off or during flight to the on board computer and can be changed according to the current scientific goals during the measurement flights. The data and the flight path can be monitored via live transfer at a temporal resolution of 1 Hz.

More information on the airplane itself can be found in Altstädter et al. (2015), and an update of the current instrumentation is provided in Bärfuss et al. (2018). Further, the UAS data derived during DACCIWA can be taken from Bärfuss et al. (2017) and more information on the meteorological data of the UAS is available in Bessardon et al. (2018). The meteorological sensor





package is displayed in Fig. 2a and the micro aethalometer is shown in Fig. 2b. The measuring devices used in the current study are explained in the following sections.

## 2.2 Meteorological Sensors for Turbulence Measurements

The meteorological sensors are mounted at the tip of the aircraft nose in order to assure an undisturbed air flow. Figure 2

displays the multi hole probe (number 1), the temperature sensors (number 2–4) and the humidity sensors (number 4 and 5).

The three dimensional wind vector ($V$) is derived via a multi hole probe manufactured by the Institute of Fluid Dynamics (Technische Universität Braunschweig, Germany; e.g., Wildmann et al., 2014). The multi hole probe is combined with an IMU/GPS system. The wind vector has a usable data rate of up to $100\,\mathrm{Hz}$ with an accuracy in the wind speed components of $\pm 0.5\,\mathrm{m\,s^{-1}}$ and in the wind direction of $\pm 10\,°$. The methods for the wind calculation are described in Bärfuss et al. (2018).

Figure 3a shows the power spectra of the wind vector components $u$, $v$ and $w$ during a horizontal flight in the altitude of $100\,\mathrm{m\,a.s.l.}$ on 2 July 2016. The signal power was compared with the Kolmogorov law for isotropic turbulence (green dotted line) and indicates that the spectral behaviour is in agreement with the expected distribution of up to a frequency of $10\,\mathrm{Hz}$.

The air temperature is calculated from three temperature sensors based on different measurement techniques. A finewire element was manufactured at the Institute of Flight Guidance with a high temporal resolution of $30\,\mathrm{Hz}$ and an accuracy better

than $\pm 0.05\,\mathrm{K}$ (see Fig. 2a, number 2). The sensor principle is comparable with a system described in Wildmann et al. (2013), but it was additionally protected with a housing against direct solar radiation, dust particles and mosquitoes. In addition, a factory calibrated capacitive sensor of type TSYS01 (Measurement Specialties, USA) was used with a resolution of around $0.3\,\mathrm{Hz}$ and a given accuracy of $\pm 0.1\,\mathrm{K}$ (see Fig. 2a, number 3). These two sensors were fused together into a long term stable reading with an error of less than $\pm 0.1\,\mathrm{K}$ and used for this study. Further, the air temperature was measured by an HMP110

(Vaisala, Finland) with lower resolution of $0.5\,\mathrm{Hz}$ and $\pm 0.2\,\mathrm{K}$ accuracy (Fig. 2a, number 4). Figure 3b represents the power spectra of the potential temperature $\theta$ after post-processing for the same horizontal flight on 2 July 2016. The signal power follows the Kolomogrov law up to the frequency of $9\,\mathrm{Hz}$.

The relative humidity is measured with a Rapid P14 Element (Innovative Sensor Technology, Switzerland) and a HMP110 (Vaisala, Finland), both based on the capacitive measuring principle (Fig. 2a, number 4 and 5). The measurement range is from

0 to $95\,\%\,\mathrm{RH}$ with an accuracy of $\pm 1.5\,\%\,\mathrm{RH}$ in a temperature range between 0 and $40\,°\mathrm{C}$ specified by the manufacturer. The response time of $10\,\mathrm{s}$ resulted after combining the humidity sensors complementary.

## 2.3 Aethalometer for Detecting the BC Mass Concentration

The one wavelength micro aethalometer (microAeth® model AE51, AethLabs, USA) is a lightweight tool for monitoring the BC mass concentration. It was implemented into the payload bay of ALADINA (see Fig. 2b). The aerosol inlet is installed

at the front of the aircraft close to the meteorological sensors (Fig. 2a, number 6). The air stream was run at a flow rate of $150\,\mathrm{ml\,min^{-1}}$ and it was dried with silica gel before reaching the AE51 inlet in order to avoid strong influences of moisture on the filter stripe of the sensor. The measurement range is $0$–$1\,\mathrm{mg\,BC\,m^{-3}}$ with a resolution of $1\,\mathrm{ng\,BC\,m^{-3}}$ specified by the manufacturer. After testing the micro aethalometer in the laboratory and in the field, the accuracy was determined to





$\pm\,0.2\,\mu g\,BC\,m^{-3}$ at a response time of 1 Hz. The effects of meteorological conditions (sensitivity to temperature and humidity changes) on the instrument's readings and artefacts were characterised on different types of airborne platforms (e.g., Ferrero et al., 2014; Ran et al., 2016; Chiliński et al., 2018) and will be further validated with the observations.

The BC mass concentration is estimated from the attenuation coefficient ($\sigma_{ATN}$) at the wavelength of 880 nm on an aerosol

loaded filter. In case of the AE51, a T60 Teflon-coated borosilicate glass fibre filter stripe is used that has to be manually changed after every measurement period. The measurement principle is based on the Lambert-Beer law and the attenuation (ATN) is defined as follows:

$$ATN = -100 \cdot ln\left(\frac{I}{I_0}\right), \tag{1}$$

where the attenuation is calculated from the ratio of the light intensity ($I$) transmitted through the loaded filter in comparison

with the initial intensity of the transmitted light ($I_0$) on an aerosol-free filter stripe. Taken from Hansen et al. (1984), the attenuation is proportional to the surface concentration of BC. Taking into account that the change in the attenuation ($\Delta ATN$) is caused by an increasing BC mass load on the filter stripe in a corresponding time interval ($\Delta t$), the BC mass concentration can be derived as follows (Saturno et al., 2017):

$$BC = \frac{\sigma_{ATN}}{\alpha_{ATN}} = \frac{A \cdot \Delta ATN}{\alpha_{ATN} \cdot Q \cdot \Delta t}. \tag{2}$$

$A$ is the filter stripe area, $\alpha_{ATN}$ is the BC mass attenuation cross section at the wavelength of 880 nm and $Q$ is the volumetric flow rate.

$\sigma_{ATN}$ was corrected due to artefacts based on the filter-based measurement technique (e.g., Weingartner et al., 2003; Virkkula et al., 2007; Collaud Coen et al., 2010; Ran et al., 2016). In addition, the micro aethalometer is sensitive to changes in temperature and humidity as described in the following section.

**2.3.1   Influence of Temperature Changes and Data Post-Processing**

As example, the impact of the temperature changes on the attenuation is shown for a flight that was performed with ALADINA on 14 July 2016 (flight ID 41 of the total measurement flights). Figure 4 presents the internal temperature from the micro aethalometer (PCBtemp) during the measurement period from 05:16 until 06:30 UTC that varied between 27 and 36 °C. In order to determine the temporal evolution of the temperature, the bit noise has to be filtered out by any smoothing algorithm. In

the laboratory tests, the influence factor of temperature changes on the BC measurements was determined to be in the order of $0.25\,\mu g\,BC\,(dT/dt)^{-1}$. Applied for the current case, the largest temperature gradients were observed between 05:16 and 05:30 UTC during the first steps of the ascent, leading to a bias (shown in BC Error, lower panel of Fig. 4) of $3\,\mu g\,BC\,m^{-3}$. For the whole study, the BC data were corrected with the internal temperature changes measured directly at the BC sensor.

During post-processing, the readings of the BC attenuation were phase shift free low pass filtered at different time scales

in 10, 30 and 60 seconds. Figure 5 shows the high variance of the attenuation signal for 1 s temporal resolution during a





measurement flight between 15:58 and 16:42 UTC on 10 July 2016. The results with the highest possible temporal resolution but still acceptable noise were obtained by a low pass filter with a span of 10 s. This method was further used for the study. The mean attenuation after 10 s low pass filtering was $1.59 \pm 0.30\,\mu g\,m^{-3}$ varying between 0.5 and $2.5\,\mu g\,m^{-3}$.

## 2.4 Model Description of COSMO-ART for DACCIWA

In this section, a short description of the model COSMO-ART (Consortium for Small-scale Modelling–Aerosols and Reactive Trace gases) and the simulation set up for the DACCIWA measurement campaign are shown. COSMO-ART is a comprehensive online coupled model system (Vogel et al., 2009) based on the operational weather forecast model COSMO (Baldauf et al., 2011). COSMO-ART includes a comprehensive chemistry module to describe the gaseous composition of the atmosphere and secondary aerosol formation. Chemical reactions are calculated with RADMKA (Regional Acid Deposition Model Version

Karlsruhe; Vogel et al., 2009), which is based on RADM2 (Regional Acid Deposition Model, Stockwell et al., 1990). Physical processes, including transport, turbulent diffusion, and dry and wet deposition are treated together with photochemistry and aerosol dynamics using the modal approach. The size distribution of aerosol within COSMO-ART is approximated by eleven log-normal distributions (modes), considering the Aitken and nucleation mode (with and without BC core), fresh BC, and the coarse mode (sea salt and mineral dust). COSMO-ART explicitly treats the ageing of BC particles transferring them from

external to internal mixtures as described in Riemer et al. (2004). For DACCIWA, the COSMO-ART model system was applied quasi-operational during the DACCIWA measurement campaign, to support the decision-making of research aircraft flight tracks within the aircraft special observing period (27 June–17 July 2016) and to derive model climatologies of the spatial distribution of the southern West African air pollution. The continuous forecasts were initiated on 8 May and were active until 31 July 2016, covering the simulation domain 25 °W–40 °E and 20 °S–35 °N with a grid mesh size of 28 km and

3 h model output. ICON forecasts were used as meteorological boundary conditions and MOZART (Model for OZone and Related chemical Tracers) as aerosol/chemistry boundaries. The simulations consider the emission of mineral dust, sea salt, biogenic volatile organic compounds, dimethyl sulphide and emissions from biomass burning and anthropogenic origin. BC is related to the anthropogenic emissions and the emissions from biomass burning. The feedbacks of the prognostic aerosol on the aerosol direct and indirect effect are not considered in these simulations.

## 2.5 Measurement Site

The measurement flights were performed at the local airfield (8 °1′N, 2 °27′E, 185 m above sea level, a.s.l.) south-west of Savè, Benin, in the period between 2 and 16 July 2016. The commune covers an area of $2\times10^4\,km^2$ with approximately 100,000 inhabitants. The Savè supersite of Karlsruhe Institute of Technology (KIT) and Université de Tolouse (Université de Toulouse III- Paul Sabatier, UPS) was installed at a distance of 4 km SW of the airfield close to Gobè (Fig. 6). An overview of the Savè

supersite and the mounted instrumentation are given in Kalthoff et al. (2018). Both measurement sites are connected with the main road RNIE2. The airfield consists of dry and sandy soils and is surrounded by agricultural land. During the experiment, 53 flights were performed on different daytimes with a total flight duration of 32 h and a total flight distance of approximately 2260 km. Within the study, 155 vertical profiles were realised up to a maximum height of 1600 m a.g.l. allowing the analysis





of ABL conditions in connection with the vertical distribution of BC. Although the Savè supersite was at a distance of 4 km, horizontal flights were in most cases directed parallel to the current wind direction measured at the supersite.

A typical flight pattern of a horizontal leg can be seen in Fig. 7. The black dashed line shows the flight track of one measurement flight on 14 July 2016. The starting point of the UAS is marked with the black dot at the latitude of 8.0171 °N and the longitude

of 2.4637 °E. Further, the wind speed is shown along four selected horizontal legs at the height of 520 m a.s.l. The wind speed varied between 7 and 11 m s$^{-1}$ (blue to red in the colour bar) and shows a dependence on the horizontal scale, as the wind speed increased in the south and in the direction of the Savè supersite.

### 2.5.1 Ceilometer and Wind Profiler

In order to retrieve information on cloud cover, cloud base height (CBH), wind speed and wind direction during the UAS

operation, ceilometer and wind profiler data were used from the Savè supersite.

The CHM15k ceilometer was employed to obtain the temporal evolution of cloud characteristics during the DACCIWA campaign (Handwerker et al., 2016). From the measurements of the attenuated backscatter coefficient profiles the CBH is determined based on a threshold method (manufacturer Lufft, personal communication, 2016). The manufacturer algorithm allows for the detection of up to three CBHs at a temporal resolution of 1 min and 15 m vertical resolution. In this study, only

the first detected CBH is shown, since the focus is on the LLC.

High-resolution information of flow conditions (wind speed and wind direction) is obtained from a sodar (for the lower part of the ABL, Wieser et al., 2016) and an ultra-high-frequency (UHF) wind profiler (above 200 m a.g.l.) measurements.

The sodar is an active remote sensing instrument, which was continuously running during the campaign. The retrieved information is based on the reflection of acoustic pulses at temperature inhomogeneities in the air with subsequent Doppler

analysis. The instrument provides profiles of horizontal wind speed and direction and the backscatter at the 30 min temporal resolution and 10 m vertical resolution between 30 and 600 m a.g.l. in altitude.

The UHF wind profiler installed at Savè by UPS is a 1274 MHz Doppler radar and it works with five beams to document the vertical structure of the atmospheric dynamics up to the middle troposphere. It allows the retrieval of the three components of the wind. The wind profiler operated continuously from 19 June to 30 July 2016 with two acquisition modes (75 and

150 m vertical resolution, respectively) in a time resolution of two minutes. This radar is used in Dione et al. (2018) for the characterisation of the low-level atmosphere dynamics during the whole DACCIWA campaign. More details on the data availability and the technical characteristics of this radar can be found in Derrien et al. (2016). In this study, the data was averaged over 15 minutes using the low mode (0-3 km) for the analysis of the wind speed and direction during the studied days.

### 2.5.2 Gas Concentrations of NO$_x$ and CO

Measurements of trace gases were taken on two separate towers. Nitrogen monoxide (NO) and nitrogen dioxide (NO$_2$) were measured on an 8 m high tower and carbon monoxide (CO) was measured separately at 3.50 m above ground level (a.g.l.). Both





towers were located at the Savè supersite, 80 m away from each other, and were generally upwind from the main neighbouring town of Savè and the power generator used for the whole instrumentation.

CO atmospheric mixing ratios were measured with a modified Model 48C-TL CO Analyzer (Thermo-environmental Instruments Inc.) with a detection limit of $12 \, ppb_v$. A dynamic dilution method by flow regulators is used for the CO calibration. The dilution is made with a commercial reference CO-N2 mix at $450 \, ppm_v$ (air liquid bottle) into zero air made by Sofnocat 423. Every 3 or 4 years, the flow regulators device is send to French Laboratoire national d'Essais for check and calibration.

NO and $NO_2$ were measured with a Model 42C-TL NO-$NO_2$-$NO_x$ (Thermo-environmental Instruments Inc.) with $0.05 \, ppb_v$ detection limit. The Model 42C-TL NO-$NO_2$-$NO_x$ was calibrated before and after the campaign by using a reference $NO_2$ air mixture, i.e. NO in $N_2$ diluted with zero air. Reference NO, and $NO_2$ were ISO 6141:2015 certified at 8.73 and 8.58 ppm for NO, before and after the campaign, respectively, and 9.28 ppm for $NO_2$, both with 5 % precision (Pacifico et al., 2018).

All data of the trace gases were sampled every 10 seconds, filtered and averaged to produce 1 and 30 minutes values. The data is accessible from Derrien et al. (2016).

## 3 Results and Discussion

In this study, an overview of the small-scale vertical variability of BC during the two week measurement period of the DAC-CIWA field experiment is presented. For this, the vertical distribution of the observed BC mass concentration is shown in relation to the structure of the ABL based on 155 vertical profiles obtained with ALADINA. Further, a particular case with the typical WAM conditions lasting from 14 until 15 July 2016 is analysed that emphasizes the dependence of BC concentration on transport processes. Simulation results of COSMO-ART are compared to the BC profiles obtained by ALADINA to achieve a better knowledge of the contributors and the possible sources of the observed BC.

### 3.1 Summary of the Vertical Variability of BC during the Experiment

Figure 8a shows the BC mass concentration, the potential temperature and the mixing ratio calculated in 20 m intervals in the lowermost 1100 m a.g.l. along 155 vertical profiles. Figure 8b represents the standard deviation of the mean profiles for corresponding parameters shown in Fig. 8a. The vertical distribution of the minimum BC mass concentration was almost not detectable in the lowermost 600 m a.g.l., but increased up to $1.64 \pm 0.2 \, \mu g \, m^{-3}$ at higher altitudes. The total maximum of $14.01 \, \mu g \, m^{-3}$ was measured in the height of 200 m a.g.l., but was only observed during one day on 15 July 2016. The mean BC mass concentration ($\overline{BC}$) varied between $1.79 \, \mu g \, m^{-3}$ and $2.42 \, \mu g \, m^{-3}$, and the standard deviation for all altitude intervals was $\pm 0.31 \, \mu g \, m^{-3}$. The vertical distribution of $\overline{BC}$ showed three dominant distinguished layers at the heights of 200, 600 and 920 m a.g.l. (Fig. 8b). The mean potential temperature ($\overline{\theta}$) was between 300.6 and 304.1 K in the vertical distribution and represented an overall stable stratification of the ABL, as the majority of the measurement flights was performed in the morning hours. The mean water vapour mixing ratio ($\overline{q}$) varied between 14.24 and $16.24 \pm 0.52 \, g \, kg^{-1}$ and decreased with altitude in the lowermost 1100 m a.g.l.





## 3.2 BC Observations during Case Study on 14–15 July 2016

The following section addresses the vertical distribution of BC measured with ALADINA on two days (14 and 15 July 2016) that were influenced by the NLLJ and free of LLC (14 July 2016), which is related to the vortex phase, and influenced by the NLLJ and LLC (15 July 2016), see Babić et al. (2019b).

An overview of the performed measurements during the case study is presented in Tab. 1. The flight time of ALADINA is given in UTC (local time = UTC +01:00). In addition, the 1 min averaged gas concentrations of $NO_x$ and CO (Derrien et al., 2016) are presented for the flight periods. The total maximum $NO_x$ concentration of 1.9 $ppb_v$ was observed in the morning between 06:41 and 08:40 UTC on 15 July 2016 simultaneously with the maximum of 259 $ppb_v$ of the CO concentration.

Backscatter data indicate that there is a well mixed layer in the afternoon of 14 July 2016, and LLC appeared temporarily
above 800 m a.g.l. (Fig. 9a). On the following day, LLCs formed below 250 m a.g.l. at about 7 UTC (Fig. 9b).

The enhanced aerosol load disappeared, which can be seen by the enhanced backscatter at an altitude up to 300 m from midnight to 7 UTC in Fig. 9b. Afterwards, the clouds lifted up to 750 m a.g.l. at 11:00 UTC and dissolved completely at 14:00 UTC in the lowermost 2 km a.g.l. In addition, a second layer of clouds built up at the height between 1050 and 2000 m a.g.l.

Wind conditions are shown in Fig. 10. The wind speed was moderate between 3 and 6 m s$^{-1}$ in the lowermost 2 km a.g.l. in
the night from 13 July until the early morning at 06:00 UTC on 14 July (Fig. 10a). The observed LLJ intensified in the course of the night and reached the maximum speed in the early morning hours. The strip of high wind speed has dissolved at midday, simultaneously with the occurrence of clouds. However, in the afternoon there is a low-level wind maximum of 6–8 m s$^{-1}$ at the height between 200 and 600 m a.g.l. that persisted for more than 12 h at the same altitude (Fig. 10b). At 07:00 UTC on 15 July, when LLC were present, the wind speed reached a maximum of 9 m s$^{-1}$ between the height of 800 and 1000 m a.g.l.
After the dissolving cloudiness at 12:00 UTC, a wind speed of 3 m s$^{-1}$ was observed. Wind speed and wind direction are a combination of the persistent monsoon flow and the maritime inflow that arrives typically in the early evening hours Dione et al. (2018); Adler et al. (2019).

Figure 11a displays three vertical profiles between the height of 100 and 800 m a.g.l. obtained with ALADINA at 06:15, 06:55 and 07:27 UTC on 14 July 2016. The ABL was stably stratified with the base of the inversion layer at 400 m a.g.l. At the
heights of 450 to 500 m a.g.l. a dry air mass was observed, as can be seen from the profile of the water vapour mixing ratio. Above the inversion layer and in accordance with the different type of air mass, an increase of BC was observed in the residual layer (RL). The total maximum BC of 2.75 µg m$^{-3}$ was observed at the height of 600 m a.g.l. in the first profile at 06:15 UTC, and the enhanced BC concentration at this altitude was still visible 1 h later. The wind direction varied between SE and SW and the vertical distribution of the wind speed showed a significant increase close to the inversion layer. At 07:27 UTC the wind
speed increased up to 12 m s$^{-1}$ between the height of 400 and 500 m a.g.l.

The structure of the ABL was well-mixed at noon and before the evening transition (Fig. 11b) leading to homogeneous conditions of $q$ and BC in the vertical distribution. The wind direction changed from SW to SE in the lowermost 200 m a.g.l. and the wind speed was constant with 5 m s$^{-1}$ in the lowermost 800 m a.g.l.





On the following day (15 July 2016) the ABL was influenced by the formation of LLC around 07:00 UTC. The same parameters as shown in the previous part are presented in Fig. 12. In this case, three vertical profiles are displayed at 05:33, 06:10 and 06:46 UTC before the LLC occurred. The UAS was operated between 100 and 900 m a.g.l. and the ABL was stable with a capping inversion layer between 300 and 400 m a.g.l. The fourth vertical profiles was performed at 08:29 UTC below the

cloud base maximum height of 320 m a.g.l. The last profile was carried out at 16:52 UTC after the dissolution of LLC below 650 m a.g.l. The ABL was well-mixed and $q$ decreased in comparison with the profiles in the morning hours. The impact of the LLC on the BC mass concentration can be clearly seen in the decline of BC with a difference of 1.5 μg m$^{-3}$ at the height of 300 m a.g.l.. An other impact of the vertical variability might be caused by the change in the wind direction to SSW during the day, so that a different type of air mass was observed. The wind speed increased close to ground level up to 5 m s$^{-1}$ but

decreased to 3.2 m s$^{-1}$ at 600 m a.g.l.

### 3.3 Comparison of ALADINA Observations with COSMO-ART Model Results

In this section, the ALADINA BC profiles are compared with the model results of COSMO-ART obtained during the case study. For the comparison it has to be considered that COSMO-ART has a grid mesh size of 28 km and the lowest 1000 m are resolved by only eleven layers. This makes a direct comparison with the airborne point observations difficult. Nevertheless,

the simulations with COSMO-ART provide added value to this study: On the one hand, BC can be separated into fresh BC, aged BC in the Aitken and aged BC in the accumulation mode, allowing to discuss potential sources of BC. On the other hand, the simulations allow to embed the local observations into larger spatial scales. Riemer et al. (2004) indicated BC ageing time scales of 8 h close to the source regions and 2 h above the source regions for daytime conditions during summer, predominantly related to the ageing via the condensation of sulphuric acid on BC particles.

Figure 13 shows the spatio-temporally collocated total BC profiles of ALADINA with 1 Hz temporal resolution and COSMO-ART for six different time intervals. In addition, the COSMO-ART results for fresh BC, aged BC (Aitken mode) and aged BC (accumulation mode) are indicated.

For 14 July 2016 (see Fig. 13a–d) a full diurnal cycle of BC profiles from 06:00 to 18:00 UTC is presented. In the morning hours the observations show increased concentrations above 500 m a.g.l. with the peak of around 3.0 μg m$^{-3}$ at 600 m a.g.l.

(Fig. 13a). This peak is well captured by the model. The best agreement is found for aged BC in the accumulation mode. However, COSMO-ART shows near-surface concentrations of up to 3 μg m$^{-3}$ that are higher than the observations. Until noon, the vertical profile of BC shows a vertically constant BC concentration, likely due to the vertical mixing in the ABL (Fig. 13b and c). COSMO-ART results of aged BC in the accumulation mode match very well these observations. The observed BC concentration peak at the altitude of 900 m is not seen by COSMO-ART, and may be an artifact induced by the formation of

cloud patches as seen by the ceilometer (Fig. 9a). Between 12:00 and 18:00 UTC, no significant change in the measured vertical BC profile is visible. In the simulations, the contribution of BC in the accumulation mode decreases and the concentration of BC in the Aitken mode increases, leading to a better agreement of the observations with the total BC concentration (Fig.13d). At 06:00 UTC on 15 July 2016 (see Fig.13e) observation and model indicate an enhanced BC concentration below 400 m. The vertical profile, capturing the lowest 900 m, agrees very well between observation and model of the total BC. Around the height





of 200 m, ALADINA indicates very high BC concentrations of more than $16\,\mu g\,m^{-3}$ that are not represented in COSMO-ART. It cannot be ruled out that this is an artifact induced by the high humidity prior to the onset of cloud formation, which occurred around that time. Finally, at 18:00 UTC, the observed BC profile agrees well with the aged BC in the accumulation mode.

Figure 14 shows a South-North cross section of the total BC concentration and wind speed up to an altitude of 1500 m a.g.l. for 14 July 2016. The location of the airborne observations at Savè is indicated by the dashed line. The wind speed transect clearly shows the NLLJ with its maximum around the height of 500 m a.g.l.. From 13 July 2016 at 21:00 UTC until 14 July 2016 at 09:00 UTC, the jet propagates northwards. The vertical transect of the BC concentration indicates that the LLJ is linked to lower BC concentrations, which is confirmed by the ALADINA observations between 06:15 and 07:30 UTC on 14 July 2016 (Fig. 11). The coastline is approximately at $6.2\,°N$, so the low BC concentrations especially occur over the Gulf of Guinea. In front of the jet, an air mass with high BC burden (up to $20\,\mu g\,m^{-3}$ near the surface) is visible. At 00:00 UTC, the BC concentration maximum is directly over Savè, however, this is not related to local emissions. Four aspects seem to justify this hypothesis: (1) The transects east and west of the shown Savè transect provide a similar pattern (not shown). Therefore it is rather a zonal symmetrical feature than a local Savè feature. (2) Fig. 13 shows negligible fractions of fresh BC which indicates less influence of local emissions. (3) Concentrations of $NO_x$ and CO (not shown) do not indicate anthropogenic local emissions for the observed time period. (4) By considering also the time steps before 13 July 2016 at 21:00 UTC, it becomes clear that the polluted air mass is transported from the South in front of the northward propagating maritime inflow. Since the LLJ has its maximum around 500 m in altitude, the clean air mass of the jet puts a wedge into the polluted air mass in front (e.g. visible on 14 July 2016 at 06:00 UTC). This leads to a near surface pollution branch below the jet axis and an elevated pollution branch above the jet axis. This is also visible in the modelled profiles of Fig. 13a, and in the observed profiles, a clearly enhanced BC concentration in the upper branch, a minimum at around 400-500 m and a slightly enhanced near-surface BC concentration is visible. At 12:00 UTC, the vertical mixing in the boundary layer leads to a rather homogeneous distribution of BC in the vertical profile. Even if the peak observed at 12:00 UTC around 900 m over Savè (see Fig. 13c) may be partly induced by cloud artifacts, its altitude fits to the location of the modelled elevated pollution branch in Fig. 14. From 14 July 2016 at 12:00 UTC on, a reestablishment of a pollution layer can be observed between $7\,°N$ and $8\,°N$.

## 4 Conclusions

This study focused on the vertical distribution of BC in the ABL obtained with the unmanned aerial system ALADINA, that was operated during the field experiment of DACCIWA between 2 and 16 July 2016. In total, 53 measurement flights were carried out up to a maximum height of 1600 m a.g.l. at the local airfield of Savè/Benin in West Africa. BC was measured with a micro aethalometer, model AE51, on board the new set up of ALADINA (Bärfuss et al., 2018). After low pass filtering the attenuation coefficient with a resolution of 10 s, BC was averaged for altitude intervals of 20 m along 155 vertical profiles for a statistical overview. In addition, the BC data was further corrected with bias caused by the depencence of the instrumentation on internal temperature during ascents and descents. It was shown that maxima and minima of BC concentration are related to horizontal advection processes, and not local emissions.

One case study (14–15 July 2016) was presented in detail, influenced by NLLJ and LLC-free on 14 July, and influenced by NLLJ and LLC on 15 July 2016.

A maximum of BC=$2.81 \pm 0.30\,\mu g\,m^{-3}$ was observed in the RL between the heights of 400 and 600 m a.g.l. in relation to the NLLJ with wind speeds larger than $12\,m\,s^{-1}$. The lifted BC layer was vertically mixed during clear sky conditions during the day and the mass concentration increased continuously in the lowermost 1100 m a.g.l. On the next day, LLC formed in the early morning that led to wash out in the lowermost 400 m a.g.l. which can be further seen in the decline of the BC mass concentration profiles from $3.87 \pm 0.79$ to $2.51 \pm 0.13\,\mu g\,m^{-3}$ in the same altitude.

In addition, gas concentrations of $NO_x$ and CO were studied for the time intervals of the ALADINA flights that did not show any local emissions influence during the case study at ground. The conditions of the vortex phase on 14 July 2016, unusual during the monsoon season, included transport of aged biomass burning aerosol and decreased humidity, as described in Knippertz et al. (2017) and Flamant et al. (2018). Therefore, when comparing the case study in Sect. 3.2 and Sect. 3.3 with the mean statistics presented in Sect. 3.1, an enhanced BC concentration can be expected, which was confirmed by the observations and simulations.

The UAS observations were supported and spatially extended by a comparison with the model output of COSMO-ART. In all cases, the contribution of fresh BC to the total BC is negligible in COSMO-ART, indicating that local sources are not the major contributor of BC over Savè. The analysis of longitudinal vertical transects of the modelled wind speed and the BC mass concentrations reveals that transport processes with maritime inflow from the south are the most relevant contributors of the observed BC.

*Data availability.* The data used in this study (Derrien et al., 2016; Handwerker et al., 2016; Wieser et al., 2016; Bärfuss et al., 2017) are available on the SEDOO database (http://baobab.sedoo.fr/DACCIWA/).

*Author contributions.* BA wrote the publication under critical review and input from all co-authors. KD and BV provided the COSMO-ART model contributions. KB delivered the ceilometer profiles. CD studied the wind profiler data. FP and CJ were responsible for the gas concentration observations. KB, FP, and AL handled the preparation and operation of the UAS, as well as the post-processing of the UAS data. FE studied the statistical analysis of the UAS measurements. BA, NK and FL coordinated the project, and provided access to ground observation data.

*Competing interests.* The authors declare that they have no conflict of interest.

*Acknowledgements.* The DACCIWA project has received funding from the European Union Seventh Framework Programme (FP7/2007-2013) under grant agreement no. 603502. BA was financially supported by the German Research Foundation (DFG) under the project number



LA 2907/5-2. The authors from the TU Braunschweig wish to thank UPS and KIT for being part of the DACCIWA project. We gratefully acknowledge Lutz Bretschneider, Endres Kathe and Birgit Zachrau for their valuable support in preparation of the UAS ALADINA and during the experiment on site. The authors acknowledge support by the German Research Foundation and the Open Access Publication Funds of the Technische Universität Braunschweig.



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



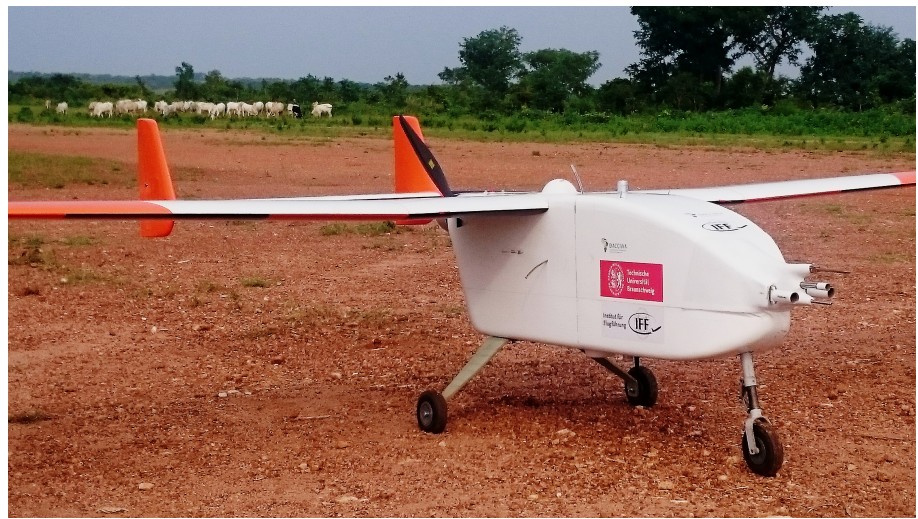

**Figure 1.** The unmanned aerial system ALADINA before take off during the DACCIWA experiment at the local airfield of Savè.
© Institute of Flight Guidance, TU Braunschweig.




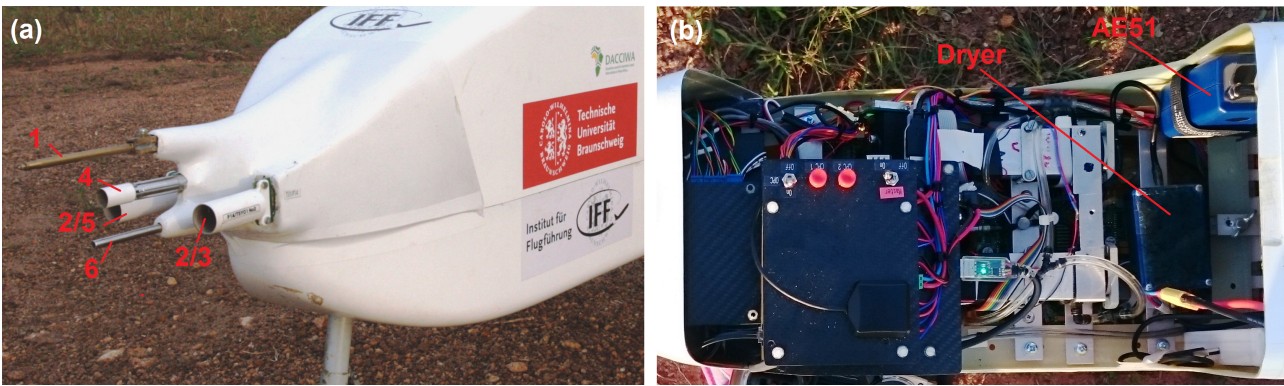

**Figure 2.** (a) The nose of the aircraft is equipped with meteorological sensors for calculating the three dimensional wind vector with one multihole probe (number 1), the temperature (number 2–4) and the humidity (number 4 and 5). The air probe is sampled with one aerosol inlet (number 6) mounted at the tip of the aircraft's nose. (b) The micro aethalometer AE51 is installed in the front compartment of the aircraft for adjusting the centre of gravity. The air stream is dried with silica gel before measuring (see the blue box close to the AE51). © Institute of Flight Guidance, TU Braunschweig.



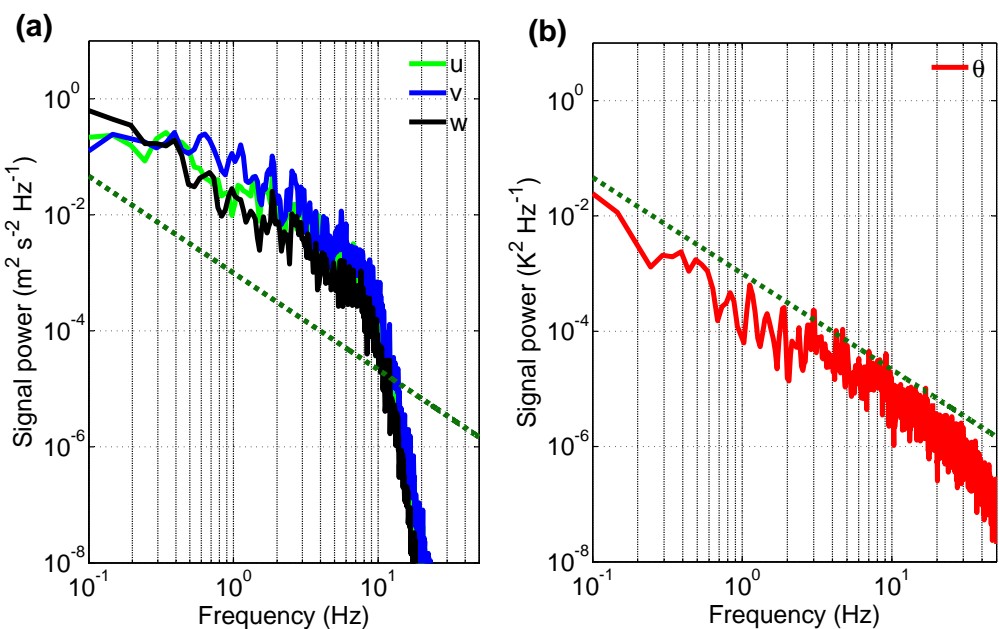

**Figure 3.** Power spectra of (a) the wind vector components *u* (green line), *v* (blue line) and *w* (black line) derived from the multi hole probe and (b) potential temperature $\theta$ (red line) after complimentary filtering of the slow and fast temperature sensors. The green dashed line represents the Kolmogorov law with a slope of $^{-5/3}$. The data was derived from a horizontal flight at the constant height of $100 \pm 1.5$ m a.s.l. on 2 July 2016.

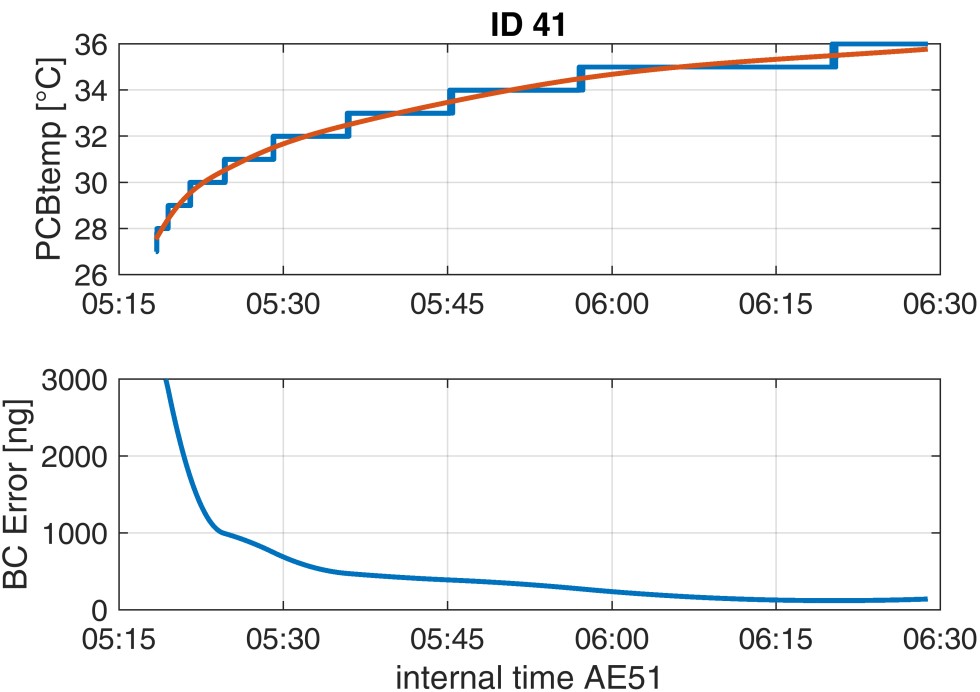

**Figure 4.** The impact of the temperature changes on the BC mass concentration. The figure shows the internal temperature (PCBtemp) of the aethalometer AE51 during the measurement flight on 14 July 2016 from 05:16 until 06:30 UTC (top). The bit noise was smoothed for the time interval (red line). The figure at the bottom represents the calculated error of the black carbon mass concentration.



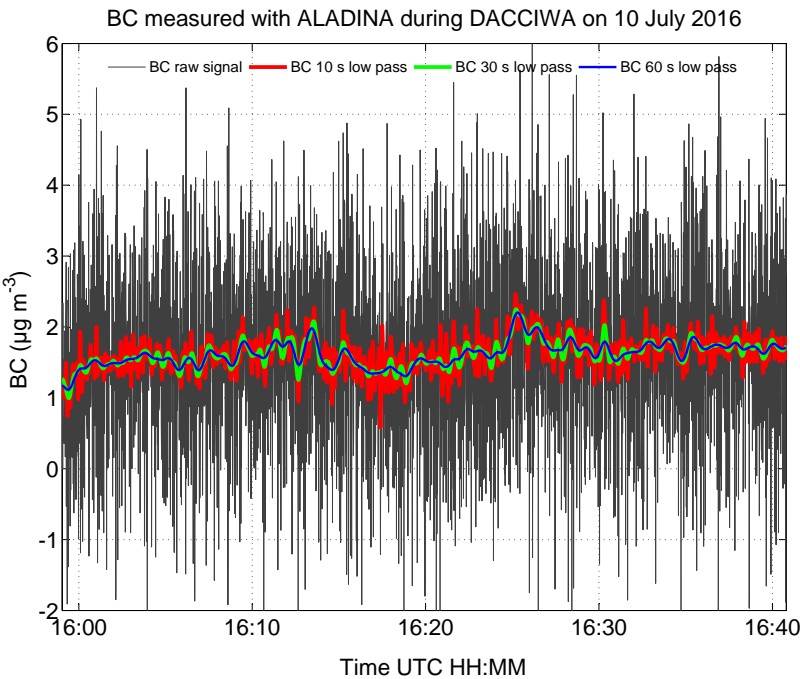

**Figure 5.** The BC mass concentration measured during one flight between 15:58 and 16:42 UTC on 10 July 2016. The raw signal of BC (grey line) was phase shift free low pass filtered after post-processing with 10 s (red line), 30 s (green line) and 60 s (blue line).



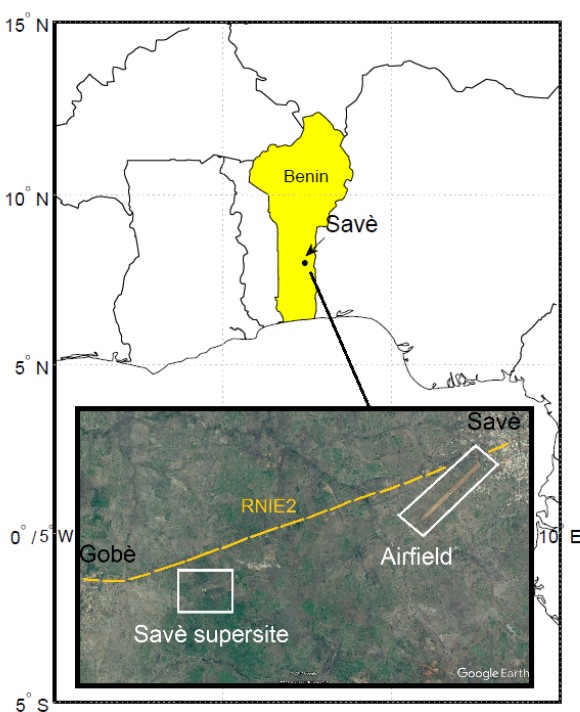

**Figure 6.** The map shows the position of the measurement site Savè in Benin (marked in yellow), West Africa. The UAS ALADINA was operated at the local airfield of Savè during the field experiment DACCIWA from 2 until 16 July 2016. The Savè supersite of Karlsruhe Institute of Technology (KIT) and Université de Tolouse III- Paul Sabatier (UPS) was located in the south-west with a distance of 4 km to the airfield. The sites are connected via the main road RNIE2 (dashed yellow line). The picture was created with © **Google Earth** on 28 April 2017.

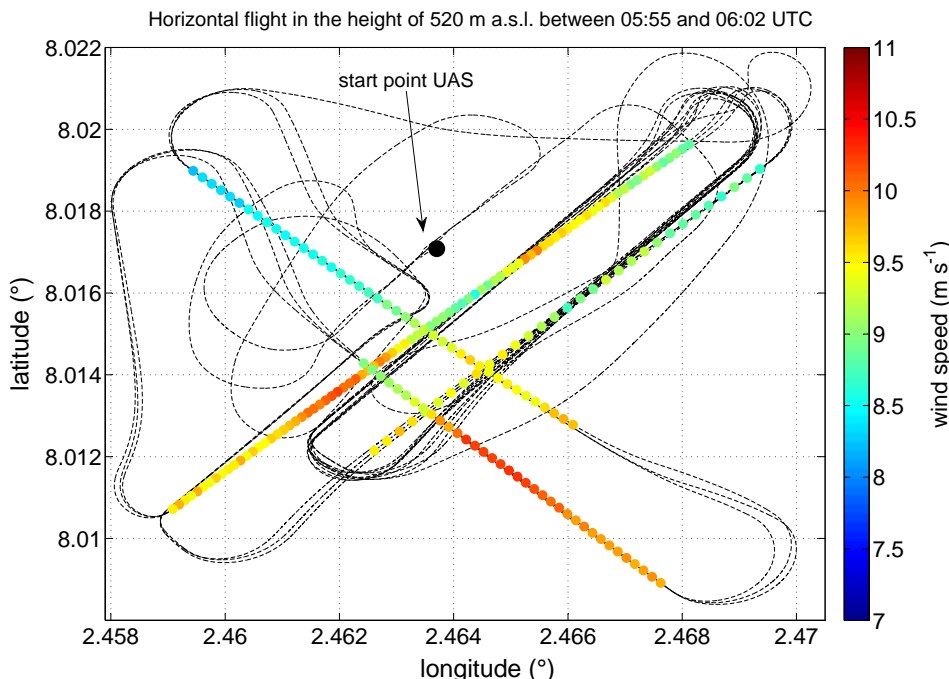

**Figure 7.** A typical flight pattern of ALADINA (black line) during DACCIWA. The start position of the UAS is marked with a black dot. The colour bar indicates the wind speed of four horizontal legs in the constant height of 520 m a.s.l on 14 July 2016. The horizontal flights were orientated towards the Savè supersite and the pattern were flown parallel and perpendicular to the current wind direction. The wind speed varied between 7 and 11 m s$^{-1}$ and shows larger wind speeds heading to the south.



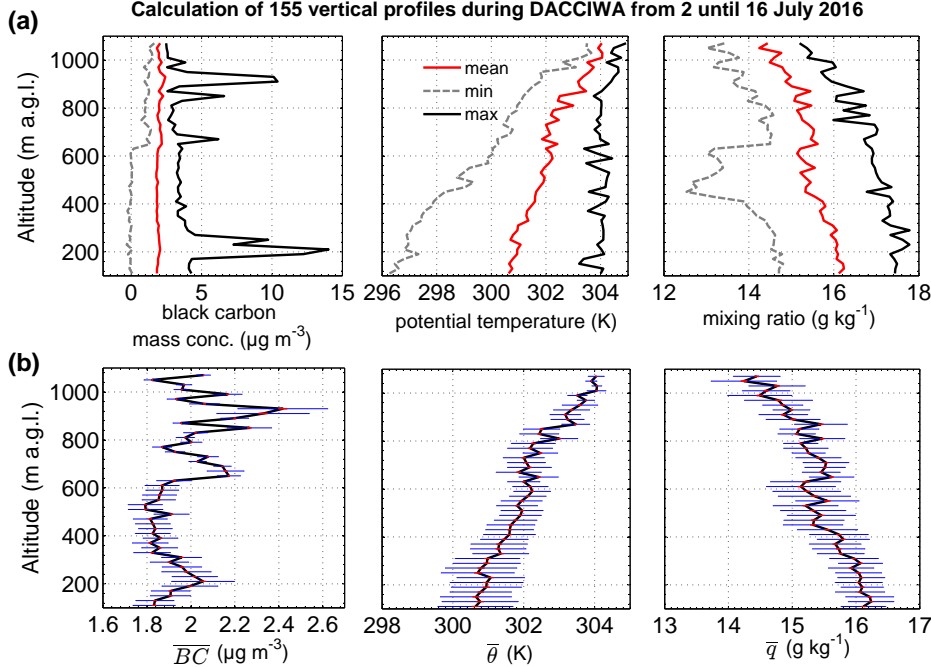

**Figure 8.** Summary of 155 vertical profiles from 100 to 1100 m a.g.l. measured with ALADINA during the field experiment of DACCIWA lasting from 2 until 16 July 2016. From left to right: Vertical profiles of black carbon mass concentration (BC) measured with the aethalometer and 10 s low pass filtering, potential temperature $\theta$ and water vapour mixing ratio $q$. The red line represents the mean value, the grey dashed lines stands for the total minimum and the black solid line is the total maximum of all vertical profiles, averaged in 20 m altitude intervals. At the bottom: The mean values of BC ($\overline{BC}$), $\theta$ ($\overline{\theta}$) and $q$ ($\overline{q}$) in the 20 m steps, and the corresponding standard deviation for each height interval.



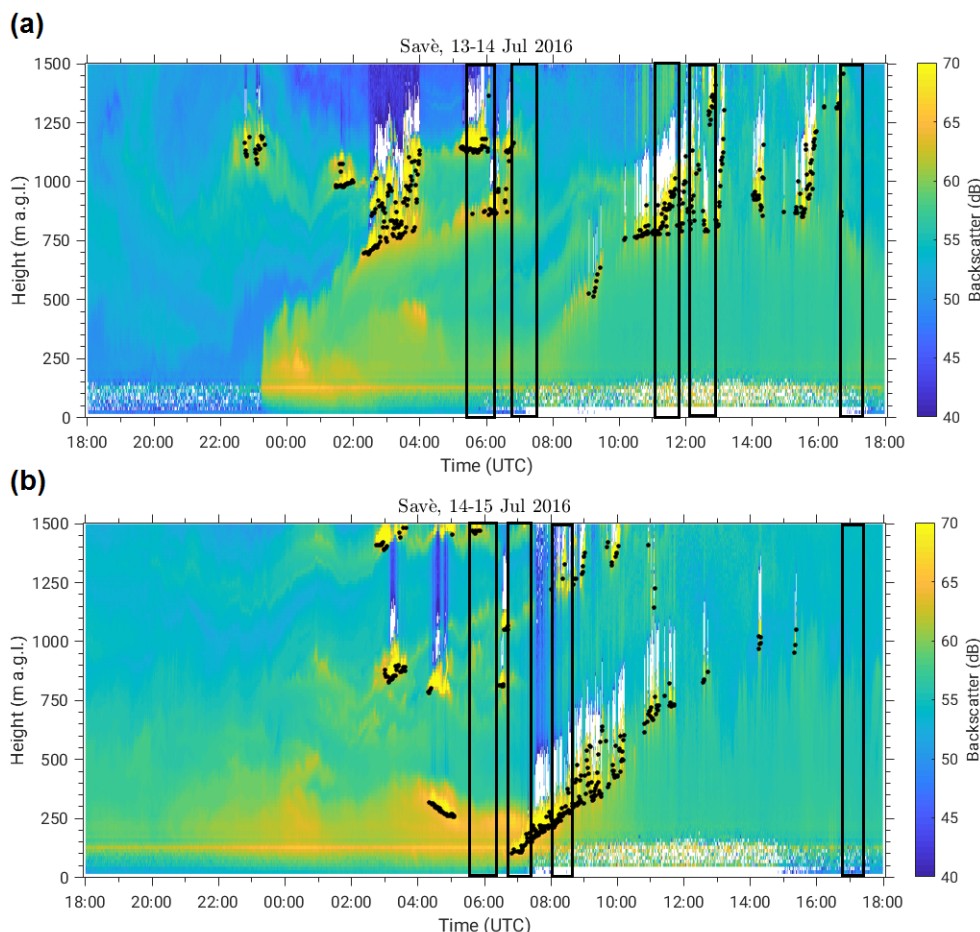

**Figure 9.** Backscatter signal of KIT ceilometer (Handwerker et al., 2016) installed at Savè supersite between (a) 18:00 UTC on 13 July 2016 and 18:00 UTC on 14 July 2016 and (b) in the period of 18:00 UTC on July 14 2016–18:00 UTC on July 15 2016. On July 14, different layers of atmospheric particles were observed during the day. At midday, clouds were detected at 800 m a.g.l. and dissolved at 18:00 UTC. On July 15, low-level clouds occurred at 07:00 UTC in the lowermost 200 m a.g.l., lifted up to the height of 750 m a.g.l. at 11:00 UTC and dissolved in the afternoon at 14:00 UTC in the lowermost 1.5 km. The black boxes indicate the flight periods of the UAS ALADINA.



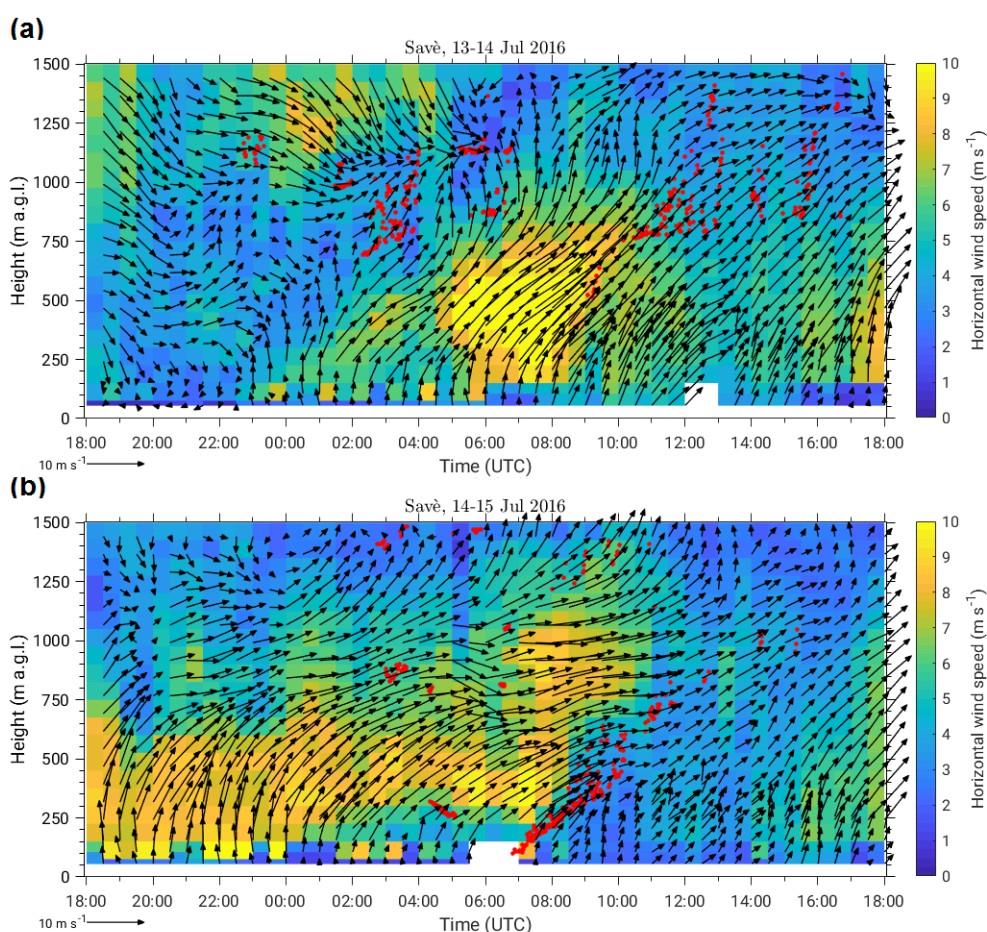

**Figure 10.** Time series of the wind speed and the wind direction at Savè supersite during DACCIWA (Derrien et al., 2016). The wind speed is indicated in the colour bar and the direction in arrows. The wind speed in the lowermost 1.5 km was measured with the wind profiler of UPS between (a) 18:00 UTC on 13 July 2016 until 18:00 UTC on 14 July 2016 and (b) from 18:00 UTC on 14 July 2016–18:00 UTC on 15 July 2016. The red dots display the cloud base height measured with the ceilometer.

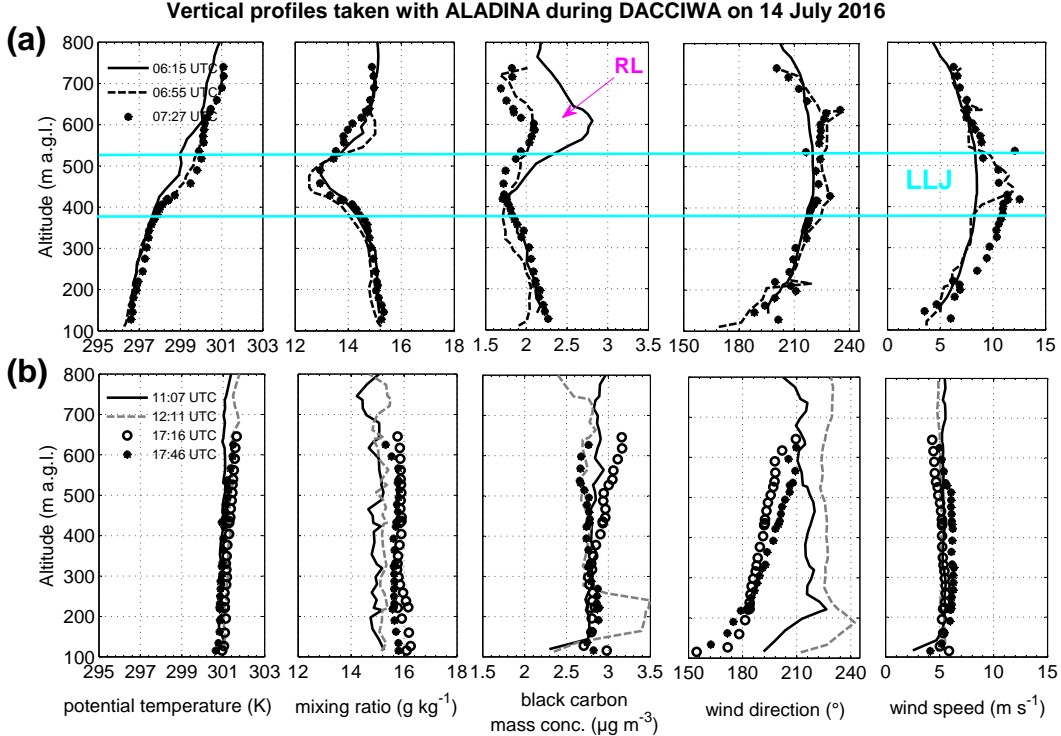

**Figure 11.** Vertical profiles taken with ALADINA between the height of 100 and 800 m a.g.l. on 14 July 2016. From left to right: potential temperature, water vapour mixing ratio, black carbon mass concentration, wind direction and wind speed (a) in the morning from 06:16 until 07:27 UTC and (b) during the cloud formation below the height of 800 m a.g.l. at 11:07 and 12:11 UTC and during the evening transition at 17:16 and 17:46 UTC. The horizontal turquoise line indicates the altitude range of the NLLJ (see Fig. 10a).

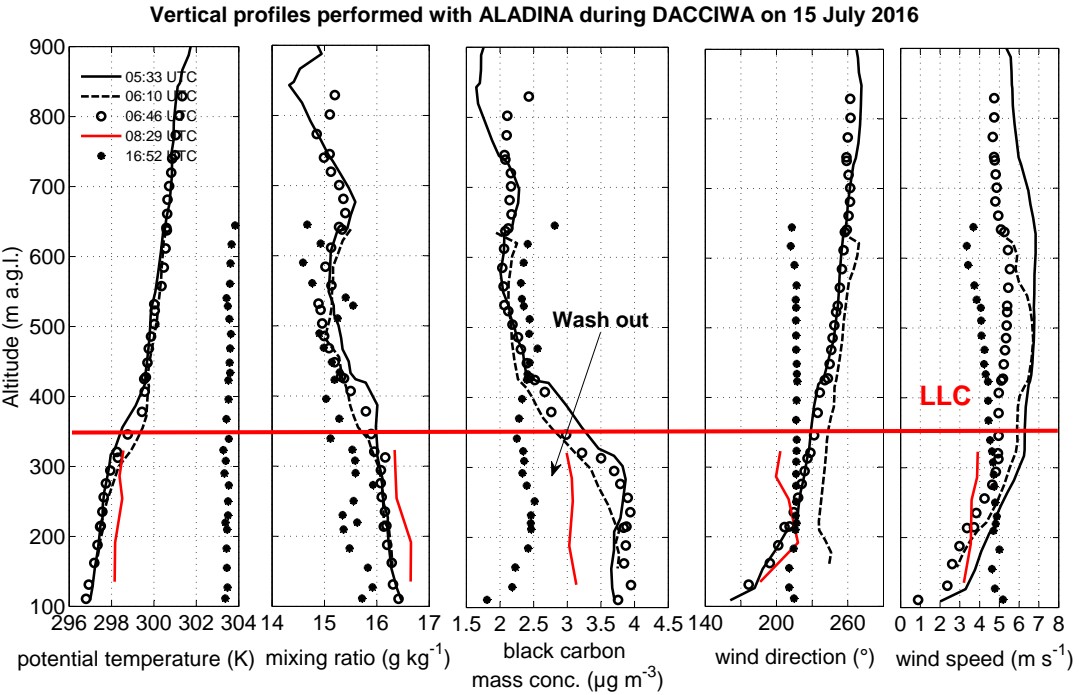

**Figure 12.** Five selected vertical profiles measured with ALADINA between the height of 100 and 900 m a.g.l. from 05:33 until 16:52 UTC on 15 July 2016. From left to right: potential temperature, water vapour mixing ratio, black carbon mass concentration, wind direction and wind speed. The red horizontal line shows the altitude of the low-level clouds (LLC) formed at 07:00 UTC, derived from the backscatter signal of the ceilometer (see Fig. 9b).

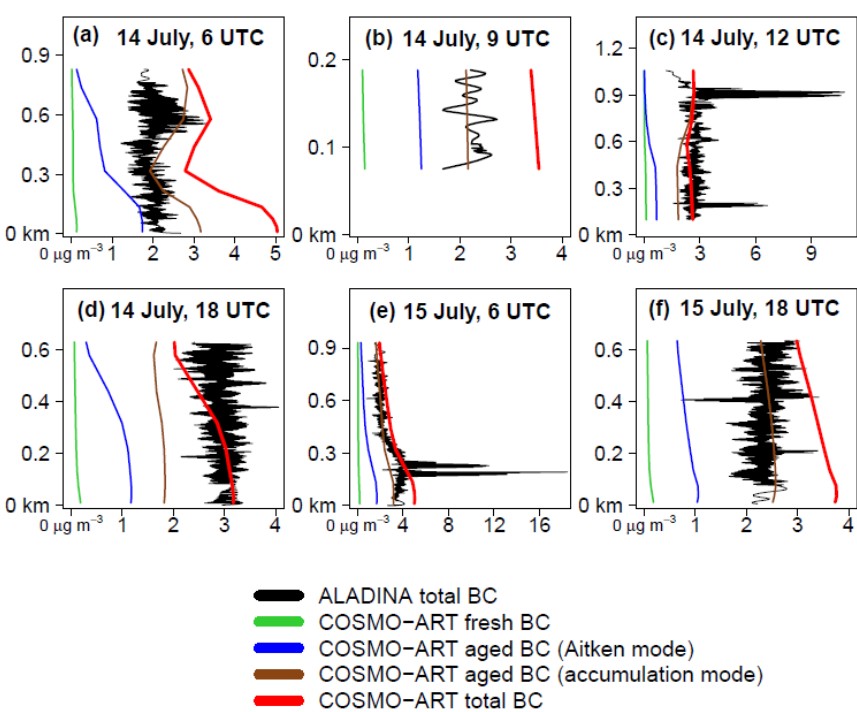

**Figure 13.** Vertical profiles of BC in µg m$^{-3}$ at Savè for (a–d) 14 July 2016 and (e–f) 15 July 2016. The ALADINA observations of the total BC are denoted in black, the COSMO-ART results for fresh BC, aged BC (Aitken mode), aged BC (accumulation mode) and the total BC are shown in green, blue, brown and red, respectively. The observations were temporally assigned to the three hourly model output with a deviation not larger than 1 h and by subsequently interpolating the model data to the ALADINA altitudes. Within these time steps, ALADINA conducted several ascents and descents. It is assumed that the observations within the time steps are measured instantaneously.

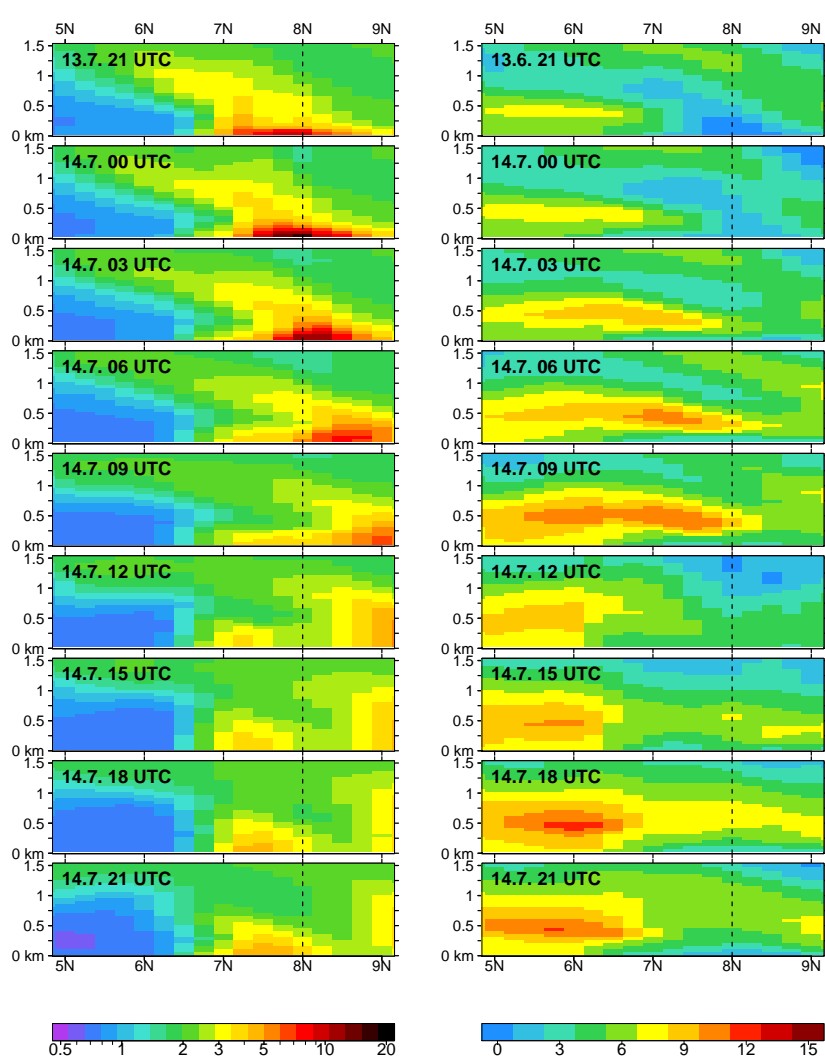

**Figure 14.** Vertical transect (km a.g.l.) from COSMO-ART of the total BC in µg m$^{-3}$ (left) and wind speed in m s$^{-1}$ (right) along the longitude of Savè between 21:00 UTC on 13 July 2016 until 21:00 UTC on 14 July 2016. The dashed line indicates the location of Savè.





**Table 1.** Measurements performed with the UAS ALADINA on 14–15 July 2016 during DACCIWA. The table presents the flight time in UTC, cloudy conditions indicated by the lowest cloud base and gas concentrations variations of $NO_x$ and CO during the corresponding flight time as an indicator of local emissions sampled at the Savè supersite by UPS.

| Flight number | Day in July 2016 | Flight time take off – landing | Cloud base | $NO_x$ | CO |
|---|---|---|---|---|---|
| 1 | 14 | 05:28–06:19 UTC | 1200 m | 0.7–1.1 ppb$_v$ | 158–231 ppb$_v$ |
| 2 | 14 | 06:51–07:32 UTC | >2000 m | 0.6–0.8 ppb$_v$ | 162–195 ppb$_v$ |
| 3 | 14 | 11:03–11:43 UTC | 1000 m | 0.7–1.8 ppb$_v$ | 153–203 ppb$_v$ |
| 4 | 14 | 12:07–12:56 UTC | 800 m | 0.6–0.8 ppb$_v$ | 146–208 ppb$_v$ |
| 5 | 14 | 16:42–17:23 UTC | 1000 m | 0.9–1.5 ppb$_v$ | 179–218 ppb$_v$ |
| 6 | 14 | 17:42–18:24 UTC | >2000 m | 1.0–1.7 ppb$_v$ | 189–232 ppb$_v$ |
| 7 | 15 | 05:27–06:13 UTC | 1700 m | 1.0–1.3 ppb$_v$ | 214–249 ppb$_v$ |
| 8 | 15 | 06:41–07:24 UTC | 300 m | 1.4–1.9 ppb$_v$ | 224–259 ppb$_v$ |
| 9 | 15 | 08:00–08:40 UTC | 400 m | 1.4–1.9 ppb$_v$ | 203–239 ppb$_v$ |
| 10 | 15 | 16:48–17:22 UTC | >2000 m | 0.8–1.4 ppb$_v$ | 156–194 ppb$_v$ |
| 11 | 15 | 17:40–18:22 UTC | >2000 m | 1.1–1.3 ppb$_v$ | 180–215 ppb$_v$ |