# Peer review of "The Vertical Variability of Black Carbon Observed in the Atmospheric Boundary Layer during DACCIWA"

_Atmospheric Chemistry and Physics, 2019_

## Referee Comment (RC1) · Anonymous Referee #1 · 29 Sep 2019

This paper describes aethalometer measurements made using a UAV in west Africa. While the measurement capability is interesting, I have some fundamental concerns with the appropriateness of the aethalometer as a measurement tool and the described data treatment. Also, the paper needs to be sharpened considerably to highlight the main messages and to put them in context so that readers can easily see the value of the findings; in my opinion it will require a major revision to be publishable in ACP. Below I include, first, my major comments and then smaller issues/recommended edits.

Broader comments: 1. The microaethelometer is not a very robust tool to measure BC. I believe that this paper serves as the first detailed description and evaluation of

the AE51 on ALADINA (the cited reference by Bärfuss shows a profile but says that the measurement is, as yet, unverified). There are some published studies evaluating its performance but, to my knowledge, not on airborne platforms, and often not in comparison to a verifiable BC measurement. The authors mention, at the bottom of page 4, that the accuracy of the microaethelometer was tested in the lab. This work should be presented fully in this paper as it speaks directly to the reliability of this data.

2. Along similar lines, I am very concerned about the temperature correction. As shown it appears to be a 3 ug/m3 correction for a change of 8C? Unless I'm misunderstanding that figure and associated discussion that is a huge correction factor. The correction is then bigger than most of the measurements. I can't even imagine what would cause the measurement to be temperature sensitive. Is it the light source? That would be a big change in intensity. Much more needs to be said about this. For example, is the case shown the worst-case scenario in terms of a flight with a strong temperature profile or is that typical? I think the authors should show the vertical profile of the correction factor. At a minimum, some discussion should be added describing the frequency with which large temperature corrections were applied and addressing the question of how errors in this correction factor might affect the interpretation presented here.

3. The comparison to the model is a bit odd. I can understand the value of comparing measured to modeled values but the way this analysis is written the model seems to be assumed to be correct rather than really evaluated. While I'm not sure that I'd call the AE51 "ground truth" I also doubt that the model is perfect and it seems like a stretch to me to rationalize model-measurement disagreement as artifacts. I'm not an expert in this field so I may not be aware of the literature but, have the emissions been fully validated? Has the aging scheme been compared to at least a handful of parallel observations? Have the modeled size distributions been validated? 20-25% of the BC mass in the Aitken mode seems like a lot and it seems like the model didn't represent the boundary layer very well (in that model concentrations seem to fall off while measurements are largely flat as a function of height). In the discussion

of local vs advected emissions, four reasons are given. Two are purely model based and not convincing (the model shows that the feature is regional, the model predicts mostly aged emissions). The third sounds reasonable (NOx and CO don't indicate local emissions) but those measurements should be shown and this should probably be the primary reason that advection is hypothesized. I don't really understand the 4th reason. Are there big cities or other BC sources between the sampling location and the coast? Otherwise I don't see how a seabreeze can bring both polluted air and clean air.

Smaller issues:

1. There are some language issues, especially towards the beginning, and I recommend English language editing. "If BC is once emitted" should be "Once BC is emitted". In the same sentence there are several unnecessary "thes", a problem which appears throughout. There are many instances where the sentence construction and grammar make it hard to understand what the authors are saying.

2. A smaller comment with regard to the data processing of the aethalometer. I don't really know what "phase shift free low pass" filtering is. The noise looks fairly random. Why not use a simple average to deal with the noisiness of the high time resolution data?

3. I'm unsure of your meaning in Line 21 with regard to hygroscopic growth. Why does hygroscopic growth enable coating?

4. There are already a lot of figures for a relatively brief paper. I don't see the value in including the power spectra graphs for the wind vectors. There you could just refer to previously published reports of that payload, especially since the present analysis doesn't seem to make use of high-resolution wind products.

5. In figure 8, I recommend that you change the x-axis label in the upper right panel to say "water mixing ratio" rather than just "mixing ratio".

6. In figure 12 you label one line as "wash out". This term is also used in the conclusions but not in the discussion. What do you mean by this? Are the low level clouds precipitating or is it just that cleaner air blew in? If it's the latter I wouldn't call that wash out, I would call it advection of cleaner air.

7. On the wind plots, how is the direction indicated by the arrows? I'm used to seeing these plotted on graphs with lat-lon as the axis but I'm not sure they make sense plotted on axes of height-time. I imagine you are just showing direction where N is "up" so perhaps the solution is to simply include a legend showing that? It's kind of an awkward graphic but I don't have a great solution for it.

8. There are a number of acronyms that are only used a handful of times. Notable NLLJ and LLC. Given the frequency of usage I recommend that, those just be written in full.

---

## Referee Comment (RC2) · Anonymous Referee #2 · 1 Oct 2019

This paper presents aethalometer measurement from the field experiment DACCIWA conducted in the atmospheric boundary layer in Benin. The campaign is interesting and conducted in a region where better observational coverage is needed, and black carbon in the boundary layer is interesting for a number of reasons. The paper hence provides a topical contribution to the field. However, in my opinion the manuscript needs substantial additional work to improve presentation, message and context.

General comments:

While results are presented clearly and in great detail, I'm struggling to identify and capture the main messages of the study. For instance, the "Conclusion" section does

not really provide any conclusions or outlooks, and the paper does not contain much in terms of discussion of relevance of results from a broader perspective.

Similarly, a bit more context in terms of why these measurements are important would be useful. E.g., the introduction could focus a bit more explicitly on the importance of BC in the boundary layer in terms of disturbances to turbulence, citing some recent studies such as Wilcox et al. (PNAS, 2016), Talukdar et al. (2019, JGR-atmospheres), Liu et al. (Atmospheric Pollution Research 2018), Gao et al. (ACP 2018).

The language needs to be reviewed and improved. There are a number of somewhat strange and incorrect sentence structures, making the manuscript difficult to follow. And a lot of unnecessary "the"s. Some of the acronyms would be better to spell out.

More specific comments: I think the authors should use the recommendations from Petzold et al. 2013 regarding terminology for reporting measurements of "black carbon" and use EBC here (or at least be clear that the term BC is used as a generic term).

On page 3, it is pointed out that the conditions during the selected cases are unusual for the monsoon period. What does this imply for the representativeness and usefulness of these results?

Suggest combining Figures 1 and 2 as the paper already contains a large number of plots.

Section 2.3.1: a brief introduction to this section about what causes this temperature effect would be very useful for non-experimentalist readers.

Page 6, line 3: how is "acceptable" defined?

Section 2.4: A brief description about the parameterizations of wet and dry removal of the aerosols would be useful.

Page 6, line 31: does the presence of dry and sandy soils present a source of uncertainty in the measurements?

Page 10, lines 15-17: please check the use of "on one hand (. . .). On the other hand (. . .)".

Page 10, lines 178-19: as the observations do not allow for an investigation of evolution of mixing state, I don't quite see why this information is included.

Page 11, line 3: this statement seems a bit odd – you'd want the observed profile to agree with total modeled BC. Individual size modes may be important for discussion of why the discrepancy with total BC is there.

References: https://www.ncbi.nlm.nih.gov/pmc/articles/PMC5081626/ https://agupubs.onlinelibrary.wiley.com/doi/abs/10.1029/2018JD029611 https://www.sciencedirect.com/science/article/pii/S1309104217306773 https://www.atmos-chem-phys.net/18/7081/2018/ https://www.atmos-chem-phys.net/13/8365/2013/acp-13-8365-2013.pdf

---

## Author Comment (AC1) · 2 Apr 2020

Please find attached the authors' response to RC1 and RC2 in a point-by-point way. The changes are marked in the annotated manuscript enclosed to this response.

Please also note the supplement to this comment: https://www.atmos-chem-phys-discuss.net/acp-2019-555/acp-2019-555-AC1-supplement.pdf

---

## Author Response (AR1)

**Authors' response to RC1 and RC2**

1

**Manuscript number: acp-2019-555**

**5 The Vertical Variability of Black Carbon Observed in the Atmospheric Boundary Layer during DACCIWA by Barbara Altstädter et al.**

First, the authors acknowledge both referees for sharing their valuable
comments! The authors agree that the goal of the study was not exactly formulated in
the previous manuscript. Please, take into account the changes in the marked-up
manuscript. The sections "Abstract" and "Conclusion" were completely revised.
Further, the new manuscript will benefit from a more detailed description of the
technical part. For instance, the explanation of "phase shift free low pass filtering"
and information on data treatment were extended. We agree that this is mandatory in
order to sufficiently describe the used methods for a broader range of ACP readers.
Before starting the responses, just some short hints in advance:

The authors will take into account the editorial support for the revised manuscript. Language issues were criticised by both referees and might have led to 20 misunderstandings in the discussion paper.

Please note that some literature was updated during review process. The current references are listed at the end of the authors' response and were added to the marked-up manuscript.

Now, find attached the authors' response according to the two referee comments (RC1 and RC2) in a point-by-point way. The changes are marked in the annotated manuscript enclosed to this response.

**Comments of referee #1 (RC1)**

"This paper describes aethalometer measurements made using a UAV in West
Africa. While the measurement capability is interesting, I have some fundamental concerns with the appropriateness of the aethalometer as a measurement tool and the described data treatment. Also, the paper needs to be sharpened considerably to highlight the main messages and to put them in context so that readers can easily see the value of the findings; in my opinion it will require a major revision to be publishable in ACP. Below I include, first, my major comments and then smaller

**Author's response to referee #1**

issues/recommended edits."

The authors thank the anonymous referee for the critical and thorough reading and for providing specific comments that will improve the current manuscript. We agree that data post-processing should be described more in detail in order to make it clear for all disciplines of ACP.

**Point-to-point response:**

20

25

**RC1**: "1. The microaethelometer is not a very robust tool to measure BC. I believe that this paper serves as the first detailed description and evaluation of the AE51 on ALADINA (the cited reference by Bärfuss shows a profile but says that the measurement is, as yet, unverified). There are some published studies evaluating its performance but, to my knowledge, not on airborne platforms, and often not in comparison to a verifiable BC measurement."

**Authors' response:**

30

The authors agree that the micro aethalometer is not the most reliable tool for detecting BC. However, it is very feasible for UAS operation, as integration of new sensors is mostly restricted by limitations of weight and size. The size of AE51 is small and the weight is not exceeding more than 300 g!

It was used for some studies on light airborne platforms like UAS and tethered balloons (e.g. Chilinski et al., 2018; Ferrero et al., 2014). To our current knowledge,

the AE51 was not tested on manned aircraft simultaneously with so called "reliable tools" (e.g. MAAP, SP2) so far. If we missed out any literature, we would be grateful for further information.

- 5 Before answering in full, the authors want to take the opportunity to explain the missing comparison of airborne BC data with other BC methods/stations/institutes that would be definitely helpful to verify the shown data. This is not foreseen as any excuse, but it would help to understand the chosen validations and the results why comparing AE51 data with filter samples and COSMO-ART model, as it is shown in the manuscript. Please note that the micro aethalometer was implemented on the 10 UAS ALADINA during DACCIWA, but it was not the main goal to study BC during the aircraft field campaign. The airborne measurements were applied for providing additional information on meteorological parameters, especially for in-situ observations performed by UPS and KIT at Savè supersite. During field study, it turned out that the AE51 worked properly on board ALADINA and there was no loss 15 of data. The large amount of 155 vertical profiles, including BC, is unique and led to further deeper analysis. After post-processing, the BC showed distinct layers in the vertical distribution, but could not be explained solely by the airborne data. The authors of TU Braunschweig decided to focus on the measured BC data, in particular
- 20

This is the reason why filter samples of Savè supersite are used, although it has to be considered that these measurement methods are different and the calculated BC is not the same (here optical equivalent BC vs chemical analyses of BC). But, it is still the best way (or let's say "a better first step") to find possible 25 sources or impacts of the clear distinct layers that were observed in the vertical distribution with ALADINA. The initial expectation was to identify any local emissions with the help of filter data. But there was no clear evidence of biomass burning in NOx or CO. Unfortunately, other BC measurements were lacking at the Savè supersite. rBC was measured with SP2 on manned aircraft (e.g. Flamant et al., 2018; 30 Brito et al., 2018) but both measurement systems are not comparable due to sampling different air masses. The flight strategy of UAS is targeting small scale studies in contrast to large scale studies by manned aircraft with a fast cruising speed.

for the case study on 14 and 15 July 2016 and were strongly supported by additional

measurements conducted at Savè supersite (4 km distance).

But, the large variability of BC in the lowermost 1 km could be better understood by the impact of the prevailing weather situation, shown with ceilometer and wind profiler. The study benefits from the comparison with COSMO-ART in order to get a broader perspective of BC transport, as all other instruments were running at Savè supersite, 4 km away from the research flights. Although the validation may look a bit odd, validated transcends definitely help to verify the source of long range transport of BC. Therefore, we decided to keep it. This may help to understand the decision of using the model output (referring to Comment RC1 3.).

However, during review process the study of Pikridas et al. (2019) was 10 published that showed the comparison of three different miniaturised absorption instruments on UAS with ground based monitoring of MAAP and AE33. The AE51, a DWP (dual-wavelength prototype of AE51) and STAP were used on board three different types of UAS. Flights were carried out at an urban location (Athens) and at a background location in Cyprus. A direct comparison of each of the instrumentation 15 was done at ground when no vertical profile was operated. In addition, a multicopter sampled between 2 and 3 min duration at ground level. The combination of samples was used for a direct- so called intercomparison- of the instrumentations. Some results of the Athens campaign: The correlation of AE51 and MAAP/AE33 was R2=0.76, probably caused by the lower signal to noise ratio in contrast to the other 20 sensors. BC mass concentrations were underestimated by 6 to 7% in relationship between MAAP and AE33. The comparison during the Cyprus campaign showed a high overestimation (22+/-55% EBC) of the AE51, possibly caused by low background concentration and low sampling flow. Interestingly, the comparison was done on a multicopter so that high vibrations of the rotor blades could probably 25 influence the sampling flow as well, but the effect was not considered in the study. Nevertheless, the study offers a great opportunity for a better understanding of the AE51 signals on board in contrast with ground observation. The authors decided to include the study in the new manuscript in order to verify the capability of the micro aethalometer. 30

We added following lines to the new manuscript (marked up manuscript p. 7, I.13-25):

"However, a direct comparison of the AE51 on UAS relating to ground observations was missing so far. Pikridas et al. (2019) showed that the AE51 is a feasible tool for

4

BC measurements on UAS in an area of high background aerosol particle concentration. Flight campaigns were conducted in Athens and Cyprus. The results are based on comparison of three different miniaturised absorption instruments (e.g. AE51) on UAS with ground based monitoring of MAAP (Multi Angle Absorption Photometer) and one micro aethalometer of type AE33. According to results of the 5 Athens campaign, the correlation of AE51 was R2=0.76 by sampling close to ground in relationship between MAAP and AE33. BC was underestimated by the AE51 of 6 to 7%, thus within the given accuracy of 10%. However, the comparison during the Cyprus campaign showed a high overestimation of 22-55% BC by the AE51, possibly caused by low background aerosol particle concentration at the research 10 site. The low performance of the instrument for small background concentration in generally clean air masses is one major issue of the AE51 (e.g., Ferrero et al., 2014; Lee, 2019). The impact should be of minor relevance for the current study that addresses an area with high PM concentration, partially higher than WHO guidelines (Adon et al., 2019). However, during laboratory tests it became apparent that 15 readings of the micro aethalometer are also sensitive to changes in temperature and humidity. This will be further addressed."

20 **RC1**: "The authors mention, at the bottom of page 4, that the accuracy of the microaethelometer was tested in the lab. This work should be presented fully in this paper as it speaks directly to the reliability of this data."

**Authors' response:**

- The authors refer to laboratory tests that were performed in a temperature 25 chamber after the campaign. The tests were done in order to characterise/understand the cause of temperature artefacts. We agree that this should be pointed out in a clearer way. It was assumed that temperature error was the main impact on the BC readings during the DACCIWA study.
- 30

Changes in the manuscript are provided in the answer to the next comment.

**RC1**: "2. Along similar lines, I am very concerned about the temperature correction. As shown it appears to be a 3 ug/m3 correction for a change of 8C? Unless I'm misunderstanding that figure and associated discussion that is a huge correction factor. The correction is then bigger than most of the measurements. I can't even imagine what would cause the measurement to be temperature sensitive. Is it the light source? That would be a big change in intensity. Much more needs to be said

- about this. For example, is the case shown the worst-case scenario in terms of a 5 flight with a strong temperature profile or is that typical? I think the authors should show the vertical profile of the correction factor. At a minimum, some discussion should be added describing the frequency with which large temperature corrections were applied and addressing the question of how errors in this correction factor might
- affect the interpretation presented here." 10

**Authors' response:**

Indeed, we were also a bit shocked about the temperature gradient error. Although we made intense tests in the climate chamber, we could not determine the exact source of the error. One assumption could be the load of the optical part (light 15 source/detector/reference measurement). Errors were reproducible, but the study of these errors would ask for its own paper, since many measurements and discussions rely on the AE51 in the range of these errors. Probably the description of data treatment is misleading or not understandable in the manuscript (old version p. 5, I. 27-28). "For the whole study, the BC data were corrected with the internal 20 temperature changes measured directly at the BC sensor."

Now, we will describe it more in detail:

The correction factor was used for the analysed BC data (all flights and all vertical profiles). The sharpest temperature gradients were observed at the beginning 25 of the A51 sampling, prior to take-off. The strong heating could be in relation to inner warming of the UAS when it was prepared for the next flight at the airfield (high incoming solar radiation).

The worst case scenario was shown in Fig. 4.; in most cases the temperature 30 gradients were significantly smaller for the rest of the study. We agree that pointing out the worst case would be at least the minimum information! However, the large temperature gradients at the beginning of the flights do not influence the results of the study. For further information: the high impact does not affect the calculated ascents/descents, shown in the vertical profiles in Fig. 11.

The take-off of the UAS is handled by the safety pilot with remote control and we exclude the first parts of the flight mission due to irregular flight sections (mostly between surface and 100 m) that could not be used for the wind measurements (inflight calibration etc). Therefore, the extreme parts were not considered, but we always took into account the bias of 0.25  $\mu$ gBC/(dT/dt)-1, calculated from the lab.

5

Nevertheless, we totally agree that the AE51 still needs more profound understanding in the field in terms of appropriateness!

10 We changed the paragraph to (marked up manuscript p. 7, l. 27- p.8, l. 10):

"As example, the impact of temperature changes on the attenuation is shown for a flight that was performed with ALADINA on 14 July 2016 (flight ID 41 of 53 total measurement flights). Figure 2 presents the internal temperature from the micro aethalometer (PCBtemp) during the measurement period from 05:16 until 06:30 UTC

- that varied between 27 and 36 °C. This flight was chosen as the worst case scenario during the study. In order to determine the temporal evolution of temperature, the bit noise has to be filtered out by any smoothing algorithm. Taken from laboratory tests in a temperature chamber, the influence factor of temperature changes on BC measurements was determined to be in the order of 0.25 μgBC (dT/dt)-1.
- 20 Applied for the current case, largest temperature gradients were observed between 05:16 and 05:30 UTC during the first steps of the ascent, leading to a bias (shown in BC Error, lower panel of Fig. 2) of 3 µg BC/m3. BC data were corrected with the internal temperature changes measured directly at the BC sensor for the whole study. Here, it should be clarified that these first steps of ascent, and similar to all
- other flights, do not directly affect the here presented analysis. These first steps correspond to take-off that is handled in remote control by the responsible pilot. However, exclusively automatic flight tracks were used in order to obtain comparable vertical profiles and neglecting horizontal patterns. One possible cause of these large temperature gradients might be internal heating of the aircraft. During landing and
- 30 preparation of the next research flight, the aircraft was exposed directly to sunlight. Another effect that could lead to this high BC error might be the load of the optical part. The error was reproducible in the laboratory, but the exact source of the error could not be determined. More tests in field studies would be mandatory."

RC1: "3. The comparison to the model is a bit odd. I can understand the value of comparing measured to modeled values but the way this analysis is written the model seems to be assumed to be correct rather than really evaluated. While I'm not sure that I'd call the AE51 "ground truth" I also doubt that the model is perfect and it 5 seems like a stretch to me to rationalize model-measurement disagreement as artifacts. I'm not an expert in this field so I may not be aware of the literature but, have the emissions been fully validated? Has the aging scheme been compared to at least a handful of parallel observations? Have the modeled size distributions been validated? 20- 25% of the BC mass in the Aitken mode seems like a lot and it seems 10 like the model didn't represent the boundary layer very well (in that model concentrations seem to fall off while measurements are largely flat as a function of height). In the discussion of local vs advected emissions, four reasons are given. Two are purely model based and not convincing (the model shows that the feature is 15 regional, the model predicts mostly aged emissions). The third sounds reasonable (NOx and CO don't indicate local emissions) but those measurements should be shown and this should probably be the primary reason that advection is hypothesized. I don't really understand the 4th reason. Are there big cities or other BC sources between the sampling location and the coast? Otherwise I don't see how a

20 seabreeze can bring both polluted air and clean air."

**Authors' response**

The intention of the authors was to understand possible sources of enhanced BC layers in the boundary layer. Indeed, neither AE51 data nor model should be said as "ground truth", but the validation benefits from the comparison in the vertical distribution. This is a very important point, as filter data is only reproducible at ground where no local emission was identified. Therefore, the data was only summarised in Tab. 1. But, a comparison of model with CO was presented in the dissertation of Deetz (2018) for the field campaign. COSMO-ART was compared with other models and with aircraft campaign during DACCIWA. As the full description of the model is not in the scope of the paper, we refer to the dissertation and another study of COSMO-ART by Deetz et al. (2018).

Further, Deroubaix et al. (2019) studied the diurnal cycle of NO and CO at Savè supersite between 1 and 7 July 2016. Low concentrations led to the assumption of

transport processes that could be explained by tracer experiments from possible city plumes near the coast. We will refer to the reference in the conclusion in order to clarify the assumption of polluted air masses. In addition, we have made a new order of three hypotheses, as number 1 was not shown in this article it was removed.

5

9

The authors added following lines to the new manuscript (marked up manuscript p. 16, l. 14-18):

"This hypothesis can be supported by tracer experiments of Deroubaix et al. (2019), who showed that Savè is partly influenced by city plumes near the coast, namely

Lomé, Accra and Cotonou due to maritime inflow from south. The modelled period was 1–7 July but the conditions of the vortex phase on 14 July 2016, unusual during the monsoon season, included transport of aged biomass burning aerosol and decreased humidity, as described in Knippertz et al. (2017) and Flamant et al. (2018) so that long range transport from city plumes might be the source of observed BC in Souè "

15 Savè."

20

**Smaller issues of RC1**

RC1: "1. There are some language issues, especially towards the beginning, and I recommend English language editing. "If BC is once emitted" should be "Once BC is
emitted". In the same sentence there are several unnecessary "thes", a problem which appears throughout. There are many instances where the sentence construction and grammar make it hard to understand what the authors are saying.

**Authors' response:**

10 Thank you very much for the hint. As described above, we will take into account English language editing for the revised manuscript. Then, the manuscript should be clearer in its aim. Unnecessary "thes" were avoided in order to simplify the reading flow.

15

**RC1**: "2. A smaller comment with regard to the data processing of the aethalometer. I don't really know what "phase shift free low pass" filtering is. The noise looks fairly random. Why not use a simple average to deal with the noisiness of the high time resolution data?"

20

**Authors' response:**

A moving average inherits a poor frequency response (poor stopband attenuation and slow roll-off). Therefore, we use a Butterworth IIR-Filter running forward and backward to eliminate phase shifts.

25

The explanation was added to the manuscript (marked up manuscript p. 8., l. 12-13): "Therefore, a high-pass Butterworth filter of third order was used, running forward and backward to eliminate phase shifts Averaging was not applied for the attenuation signal in avoidance of poor frequency responses."

30

**RC1**: "3. I'm unsure of your meaning in Line 21 with regard to hygroscopic growth. Why does hygroscopic growth enable coating?" Authors' response:

The authors agree that this is a really misleading sentence. The intention was to emphasise the role of BC after a long residence time in the atmosphere that is ongoing with a hygroscopic growth and the CCN activity is increased by a possible coating with other species. The sentence is rephrased and simplified.

5

The sentence is now (marked up manuscript p. 2, l. 5-7): "Aged BC can act as cloud condensation nuclei after hygroscopic growth in the atmosphere (Zhang et al., 2008), which ultimately contributes to the indirect aerosol effect".

10

15

**RC1**: "4. There are already a lot of figures for a relatively brief paper. I don't see the value in including the power spectra graphs for the wind vectors. There you could just refer to previously published reports of that payload, especially since the present analysis doesn't seem to make use of high-resolution wind products."

**Authors' response:**

Thank you a lot for the hint. We agree that the publication involves a lot of figures and the first part might distract from the results. In order to focus more on the results, the authors will remove Fig. 1 and Fig. 3. Figure 2a displays the measurement unit, including the meteorological sensors and Fig.2b shows the integration of the AE51 in the UAS. As these instrumentations were used for the study, it would not be mandatory to show the aircraft itself. We will only refer to previous publications of Altstädter et al. (2015) and Bärfuss et al. (2019). Turbulence was not calculated for the shown BC study, so that Fig. 3 will be withdrawn, as well.

**RC1**: "5. In figure 8, I recommend that you change the x-axis label in the upper right panel to say "water mixing ratio" rather than just "mixing ratio"."

30

**Authors' response:**

Thank you, this is an important issue. The authors will change the label of the x-axis, accordingly. This will be done for Fig. 8, Fig. 11 and Fig. 12 of the old manuscript, as well.

**RC1**: "6. In figure 12 you label one line as "wash out". This term is also used in the conclusions but not in the discussion. What do you mean by this? Are the low level clouds precipitating or is it just that cleaner air blew in? If it's the latter I wouldn't call that wash out, I would call it advection of cleaner air."

5

**Authors' response:**

The term "wash out" was removed from Fig. 12, as it might have led to misleading interpretation. The authors' intention was to highlight the decreasing BC in the lowermost 350 m after the onset of low-level clouds. We do not have any evidence, if the cause of the removal was wash out or just the transport of a generally cleaner air mass.

10

25

30

RC1: "7. On the wind plots, how is the direction indicated by the arrows? I'm used to seeing these plotted on graphs with lat-lon as the axis but I'm not sure they make sense plotted on axes of height-time. I imagine you are just showing direction where N is "up" so perhaps the solution is to simply include a legend showing that? It's kind of an awkward graphic but I don't have a great solution for it."

**20 Authors' response:**

Thank you for the hint. We should add more information on wind arrows. A shown leftwards horizontal arrow presents a wind direction from E. An arrow from bottom to top stands for S wind. We decided not to change the figure, but to include the information in the capture of Fig. 10, like it was done in other results of wind profiler measurements during DACCIWA (e.g. Dione et al., 2019).

**RC1**: "8. There are a number of acronyms that are only used a handful of times. Notable NLLJ and LLC. Given the frequency of usage I recommend that, those just be written in full."

**Authors' response:**

Ok thank you for the hint. The authors decided to use the full expressions for those which are only used for a few times.

**Author's response to referee #2**

**Point-to-point response:**

- 5 **RC2:** "This paper presents aethalometer measurement from the field experiment DACCIWA conducted in the atmospheric boundary layer in Benin. The campaign is interesting and conducted in a region where better observational coverage is needed, and black carbon in the boundary layer is interesting for a number of reasons. The paper hence provides a topical contribution to the field. However, in my opinion the
- 10 manuscript needs substantial additional work to improve presentation, message and context. General comments: While results are presented clearly and in great detail, I'm struggling to identify and capture the main messages of the study. For instance, the "Conclusion" section does not really provide any conclusions or outlooks, and the paper does not contain much in terms of discussion of relevance of results from a
- 15 broader perspective."

Similarly, a bit more context in terms of why these measurements are important would be useful. E.g., the introduction could focus a bit more explicitly on the importance of BC in the boundary layer in terms of disturbances to turbulence, citing
some recent studies such as Wilcox et al. (PNAS, 2016), Talukdar et al. (2019, JGR-atmospheres), Liu et al. (Atmospheric Pollution Research 2018), Gao et al. (ACP 2018)."

**Authors' response:**

The authors thank the anonymous referee for proof reading and for the specific comments that will improve the current manuscript. We agree that the previous manuscript did not point out clear enough the main massage of the study. The goal is to link between surface observations and airborne measurements within the boundary layer in the West African monsoon area, supporting the aim of DACCIWA. In this study, surface observations were used for a characterisation of gas concentrations and the upper part of the atmosphere was described with the help of wind profiler and ceilometer data.

Please see the answers at p.1. We have fully revised the abstract and conclusions to point out the main goal of study.

The abstract is now (marked up manuscript p. 1, I. 1-23):

[revised manuscript text omitted]

Thank you very much for providing the current literature! We included the study of Wilcox et al. (2016) and Liu et al. (2018) in the introduction. The references allow a more concrete analysis of the ABL height and its interaction with BC.

Following lines were added to the new manuscript (marked up manuscript p. 3, l. 5-35):

30 "However, previous data showed only the large scale variability of BC instead of columns on smaller scales of a few square kilometres above one concrete measurement field. In particular the relationship between the ABL and BC distributions needs more profound investigations. Liu et al. (2018) studied the ABL height for 347 days depending on the classification of polluted (BC>5 µg/m3) and

non-polluted events (BC<5 µg/m3) in Wuhan, China. The ABL height was suppressed on polluted days, which ultimately leads to poorer air quality. The reduction of the ABL height can be explained by the increase of absorption due to enhanced particulate matter originating from pollution (e.g., Petäjä et al., 2016). The growing absorption heats up the upper part of the ABL which further results in an increase of stability. As cause of weaker turbulence and mixing in more stable conditions, the ABL height decreases. But in most investigations, BC is solely measured at ground. Further, Liu et al. (2018) calculated the top of the ABL from lidar measurements neglecting cases with low clouds. So that studies are missing with non-clear sky days

10 like during monsoon seasons.

Therefore, a large benefit can be expected by using unmanned aerial systems (UAS) for investigating the small scale variability of BC in the ABL. Relating to BC measurements, optical methods based on light-absorbing principle are common. The development of miniaturised BC instrumentation is essential, as reduced size and

- 15 limitation of weight are the main challenges of new sensor integration on airborne platforms. Wilcox et al. (2016) presented vertical profiles of BC, measured with a miniaturised three wavelength absorption photometer (Corrigan et al., 2008) on UAS, during the winter monsoon over the northern Indian Ocean. BC loads were higher in the lowermost 3 km on polluted days in comparison with days of minor pollution
- 20 observed at the surface. A lower ABL height was estimated during polluted days, supporting the current understanding of feedback mechanisms, like it was summarised in Petäjä et al. (2016). However, the observations showed an increase of humidity in the surface mixed layer that might favour cloud formation that is in contrast to the understanding of a lower instability and minor turbulence due to absorption of aerosols in the ABL. Thus, further studies are essential to better understand the rale of PC in the ABL and to consider potentially other impacts on the
- understand the role of BC in the ABL and to consider potentially other impacts on the vertical distribution in a broader range."
- 30 **RC2:** "The language needs to be reviewed and improved. There are a number of somewhat strange and incorrect sentence structures, making the manuscript difficult to follow. And a lot of unnecessary "the"s. Some of the acronyms would be better to spell out."

**Authors' response:**

The authors agree that the language should be revised in order to avoid any misleading interpretations. Unnecessary "thes" were removed and the acronyms are provided in full. Please, see comments above on the first page and the answers to the first referee.

**RC2:** "More specific comments: I think the authors should use the recommendations from Petzold et al. 2013 regarding terminology for reporting measurements of "black carbon" and use EBC here (or at least be clear that the term BC is used as a generic term)."

**Authors' response:**

We are grateful for your recommendation. The comparison of different measurement methods is still an important issue, as the BC strongly differs in its properties. We have added the following lines in the aethelometer section. We decided not to refer to the recommendation in the introduction, as the introduction presents the role of BC in general.

- We made following change (marked up manuscript p. 6., l. 14-21): "Further, it has to be considered that different measurement methods (for instance optical or chemical) lead to different types of BC. A clear terminology is mandatory (Petzold et al., 2013). In terms of the presented study, the equivalent black carbon (EBC) was calculated from the light-absorbing aethalometer. Only for the sake of simplicity the term BC will be used hereafter."
- simplicity, the term BC will be used hereafter."

**RC2:** "On page 3, it is pointed out that the conditions during the selected cases are unusual for the monsoon period. What does this imply for the representativeness and usefulness of these results?"

**Authors' response:**

30

In our opinion, the unusual monsoon period affected the long range transport of BC. Decreased humidity might have caused a degradation of wet deposition.

5

Nevertheless, the comment is not helpful in the introduction and it was postponed to the conclusion.

5 **RC2:** "Suggest combining Figures 1 and 2 as the paper already contains a large number of plots."

**Authors' response:**

Thank you very much for the hint. We have removed Fig. 1 and Fig. 3 in order to reduce the amount of figures. Figure 2 is sufficient, as we only focus on the measurement unit. Figure 3 was withdrawn, because we did not use turbulence data for the study.

15 **RC2:** "Section 2.3.1: a brief introduction to this section about what causes this temperature effect would be very useful for non-experimentalist readers."

**Authors' response:**

Yes, indeed. We have added the information on temperature gradient error. We assume the load of the optical part for a possible cause. However, although the error was reproducible in the lab, the exact source of the error could not be determined; more tests in field studies would be mandatory.

25 RC2: "Page 6, line 3: how is "acceptable" defined?"

Authors' response: The frequency response will be lost by a span of 30 sec or more. Thus, 10 sec will still reproduce the high noise of the attenuation signal in order to capture possible enhanced BC loads in the vertical distribution. In the shown case,

30

the term of "acceptable" noise refers to standard deviation smaller than 0.3  $\mu$ g/m3 (22 %) of the signal.

**RC2:** "Section 2.4: A brief description about the parameterizations of wet and dry removal of the aerosols would be useful."

**Authors' response:**

- 5 It is beyond the scope of this publication to give detailed information on numerical simulations. The parameterizations of processes are provided in Deetz (2018) and Deetz et al. (2018).
- 10 **RC2:** "Page 6, line 31: does the presence of dry and sandy soils present a source of uncertainty in the measurements?"

**Authors' response:**

The presence of dry and sandy soil was a major issue due to possible damages during take-off and landing for the meteorological sensors, as they are mounted at the tip of the aircraft's nose. However, the housings over the finewiresensor were reliable and none of the sensors had to be replaced. The impact of dust particles on the airstream could definitely affect the sampled flow. However, no tests were performed at the measurement site. We can only guess if the filter load could be affected by the dust particles. In order to avoid any contamination during test procedures at ground, the sample flow was filtered at the inlet. In our opinion, the

procedures at ground, the sample now was intered at the inlet. In our opinion, the impact of dust particles that could be entering the inlet during take-off should be negligible in the attenuation signal (due to the scattering properties of dust) but could have led to stronger loading on the optical part. Dust particles present continuously on the filter should not affect the measurements, as the incremental changes of the optical properties are used to derive the BC mass concentration.

**RC2:** "Page 10, lines 15-17: please check the use of "on one hand (. . .). On the other hand

(. . .)". **RC2:** "Page 10, lines 178-19: as the observations do not allow for an investigation of evolution

of mixing state, I don't quite see why this information is included."

**Authors' response:**

The authors combined both remarks of the second referee hereafter. We decided to change this paragraph about the model, as the mixing of BC is not relevant for the aethalometer measurements of the shown study. The initial assumption of the sharp gradients of the measured BC in the vertical distribution was the observation of activated BC, as the flights were performed near clouds or close to the edge of clouds. However, we assume that turbulence near cloud base and probably cloud droplets disturbed the inflow and led to high BC artefacts. Nevertheless, the model still provides helpful information on the source of the aged BC properties.

**RC2:** "Page 11, line 3: this statement seems a bit odd – you'd want the observed profile to agree with total modeled BC. Individual size modes may be important for discussion of why the discrepancy with total BC is there."

Authors' response: Thank you very much. The intention was not to state which of the methods is the right one. The sentence was rephrased and possible explanations were added to the text.

20

15

The authors made this change (marked up manuscript p. 14, l. 4-10):

[revised manuscript text omitted]

**15**

25

5

After changing the discussion and conclusion, some literature was removed. To the current knowledge of the authors, the study of Bessardon et al. (2018) was not resubmitted so far, so that it was withdrawn as well in the new manuscript.

- Adler, B., Kalthoff, N., and Gantner, L.: Nocturnal low-level clouds over southern West Africa analysed using high-resolution simulations, Atmos. Chem. Phys., 17, 899–910, doi:10.5194/acp-17-899-2017, 2017.
  - Bessardon, G., Brooks, B., Abiye, O., Adler, B., Ajao, A., Ajileye, O., Altstädter, B., Amekudzi, L. K., Aryee, J. N. A., Atiah, W. A., Ayoola, M.,
  - Babi'c, K., Bärfuss, K., Bezombes, Y., Bret, G., Brilouet, P.-E., Cayle-Aethelhard, F., Danuor, S., Delon, C., Derrien, S., Dione, C., Durand, P., Fosu-Amankwah, K., Gabella, O., Groves, J., Handwerker, J., Kalthoff, N., Kohler, M., Kunka, N., Jambert, C., Jegede,
    - G., Lampert, A., Leclercq, J., Lohou, F., Lothon, M., Medina, P., Pätzold, F.,
- Pedruzo Bagazgoitia, X., Reinares, I., Sharpe, S., Smith, V., Sunmonu, L. A.,
   Tan, N., andWieser, A.: A dataset of the 2016 monsoon season meteorology
   in southernWest Africa an overview from the DACCIWA campaign, Sci.
   Data, in review, 2018.

- Gounou, A., Guichard, F., and Couvreux, F.: Observations of diurnal cycles over a West African meridional transect: Pre-monsoon and full-monsoon seasons, Bound.-Layer Meteor., 144, 329–357, doi:10.1007/s10546-012-9723-8, 2012.
- Hannak, L., Knippertz, P., Fink. A. H., Kniffka, A., and Pante, G.:Why Do
   Global Climate Models Struggle to Represent Low-Level Clouds in the West
   African Summer Monsoon?, J. Climate, 30, 1665–1687, doi:10.1175/JCLI-D 16-0451.1, 2017.
  - Hansen, J., Sato, M., and Ruedy, R.: Radiative forcing and climate response,
  - J. Geophys. Res., 102, 6831–6864, doi:10.1029/96JD03436, 20, 1997.
  - Knippertz, P., Fink, A. H., Schuster, R., Trentmann, J., Schrage, J. M., and Yorke, C.: Ultra-low clouds over the southern West African monsoon region, Geophys. Res. Lett., 38, L21808, 1–7, doi: 10.1029/2011GL049278, 2011.

15

10

The statistical results were added by following lines in order to describe the possible impact of BC in the ABL (marked up manuscript p. 11, l. 18-31).

"All mean values showed a sharp gradient at the height of 600 ma.g.l. One explanation for this could be less data availability above this height. The research flights are restricted below cloud base, as ALADINA is not rain proof. The first part of the measurement period (2-8 July 2016) was during the post-onset phase of the WAM season (Knippertz et al., 2017). This period was influenced by low-level clouds with a median height of cloud top at 587 ma.g.l., taken from 06:00 UTC radiosondes at Savè (Kalthoff et al., 2018). This part corresponds to almost half (25 of 53) of the

research flights. Although, nocturnal low-level clouds dissolved during the day, a large portion of vertical profiles still remained below 600ma.g.l. Drier periods occurred during the vortex phase of the WAM that was predominant from 9 until 17 July 2016 (Knippertz et al., 2017). Taken from the statistic overview of BC, BC seems to occur in the whole ABL. No clear evidence is visible of local pollution near surface. One explanation could be the performance of ALADINA during daytime so that in some cases the ABL was already well mixed and BC was lifted from ground through the ABL. However, this can not be supported by mean theta that shows an overall stable ABL. This relationship presumes that observed BC originated prior to the observation periods, probably during night or even earlier. Another possibility could be horizontal

advection that caused distinguish layers of BC in the ABL. Here, the understanding of BC in the ABL can not be fully described by previous observations like Wilcox et al. (2016) and Liu et al. (2018). Directly emitted BC would have probably warmed the ABL by absorption that further caused stable conditions and a lower ABL height but with increased humidity. This is in contrast to here observed reduced humidity due to the decline of mean q in the ABL. Therefore, it is essential to take into account other

The paragraph of using UAS for ABL studies was removed from the introduction to the ALADINA section (marked up manuscript p. 4, I. 21-27):

aspects that might affect the BC distribution within the ABL."

"There is a wide range of airborne platforms, including UAS operating with batteries and fuel engines, both fixed wings and multicopters that are applied for atmospheric research on a various sampling locations (e.g., Mayer et al., 2010; Marino et al., 2015; Renard et al., 2016; Jiménez et al., 2016; Cuxart et al., 2016; Båserud et al.,

15 2016; Brosy et al., 2017). The different systems have advantages and shortcomings. In terms of ALADINA, the aircraft is electrically powered, so that any contamination of the measured air probe can be prevented. Thus, a qualitative analysis of aerosol particles is guaranteed by sampling in short periods of time, like it was done in previous studies of ALADINA (Platis et al., 2016; Altstädter et al., 2018) in Melpitz,

20 Germany."

Additional information of instrumentation on ALADINA was provided (marked up manuscript p. 5, l. 10-14):

"Other aerosol sensors (optical particle counter and condensation particle counter) that are usually on ALADINA did not work properly during DACCIWA. One explanation could be damage of optical parts after transportation, but adjusting and calibration was not possible at the research site due to missing laboratory. Thus, the current study focuses explicitly on meteorology and BC and the instrumentations are explained in the following sections."

30

26

5

**The Vertical Variability of Black Carbon Observed in the Atmospheric Boundary Layer during DACCIWA**

Barbara Altstädter1, Konrad Deetz2, Bernhard Vogel2, Karmen Babić2, Cheikh Dione2, Federica Pacifico3, Corinne Jambert3, Friederike Ebus1, Konrad Bärfuss1, Falk Pätzold1, Astrid Lampert1, Bianca Adler2, Norbert Kalthoff2, and Fabienne Lohou3

[revised manuscript text omitted]

---

## Author Response (AR2)

**Authors' response to report #1 and report #2**

**Manuscript number: acp-2019-555**

**The Vertical Variability of Black Carbon Observed in the Atmospheric Boundary Layer during DACCIWA by Barbara Altstädter et al.**

The authors first of all wish to thank both referees and Dominick Spracklen for their critical input. Thank you a lot for your support by sending your comments! After revision of the new manuscript, there will be a typesetting by ACP. Thus, the authors are confident that English language will be improved thereafter. Nevertheless, some

15 grammar issues and typos are clarified now.

Subsequently, you will find the authors' response in a point-by-point way and the annotated manuscript. Please take into account further changes that were done by the authors apart from referees' remarks (see additional changes by the authors AC). This includes minor points that do not change the manuscript per se, like

20 additional affiliations of co-authors, typos and some rephrases of sentences.

**Referee report #1 (R1)**

25 *"A minor point to correct: line 8 on page 13 reads "This article aims at understanding the relationship between the vertical distribution of BC in the ABL in southern West-Africa." but seems to be missing something. Relationship between BC and what?"*

**Author's response to referee #1**

Yes, indeed, the sentence seems incomplete. Thank you!
We made following change (marked up manuscript p. 13, l. 24-25):

**This article aims at understanding the relationship between the vertical**
35 **distribution of BC and ABL processes in southern West-Africa.**

**Referee report #2 (R2)**

40

*"The authors have addressed most reviewer comments and have done a good job of making clear the limitations of the present data set. The paper will still require some english language editing and I still think there are too many figures which are not all essential to the story. Finally, although the main motivation and conclusions are*
45 *clearer now than they were in the previous version, I still think the message gets lost in the extremely detailed descriptions of individual profiles."*

The authors thank the anonymous referee for valuable support during review process. We agree that the manuscript contains a lot of figures but this is mainly
5  based on the huge diversity of measurement methods. This is even further enlarged by comparison with model description and results. In our opinion, the descriptions of each system and large explanation of specific results are essential in order to fully understand the shown data. However, we decided to do two fundamental changes in order to uniform the outcome of the article.

1) First, the authors decided to withdraw Fig. 5 that is on minor relevance for the study. The figure shows a typical flight path of ALADINA that would be mandatory for turbulence study but this is not in the scope of the current article.
15  Thus, we deleted Fig. 5 and its description from the previous manuscript (see marked up manuscript p. 8, l. 11-16).

"A typical flight pattern of a horizontal leg can be seen in Fig. 5. The black dashed line shows the flight track of one measurement flight on 14 July 2016. The starting
20  point of the UAS is marked with the black dot at the latitude of 8.0171 'N and the longitude of 2.4637 'E. Further, the wind speed is shown along four selected horizontal legs at the height of 520 m a.s.l. The wind speed varied between 7 and 11m s$^{-1}$ (blue to red in the colour bar) and shows a dependence on the horizontal scale, as the wind speed increased in the south and in the direction of the Savè
25  supersite."

2) In addition, the authors decided to change the overview in the very beginning of Sect. 3 "Results and Discussion". This implies a link of used methods to better understand the relevance for the presented case study that should be
30  reproducible for other days.

The paragraph was rephrased to (marked up manuscript p. 9, l. 22-30):

**First, an overview is presented with respect to the small-scale vertical**
35  **variability of BC that was measured with ALADINA during the two week period (2-16 July 2016) of DACCIWA. Further, a particular case (14-15 July 2016) is analysed in detail that shows enhanced layers of BC which are not issued from primary emissions. This is subsequently supported by comparison with ground data of gas concentrations and model results of COSMO-ART. During**
40  **the case study, weather conditions were typical for the WAM, mainly influenced by the occurrence of a low-level jet and low-level clouds which are shown by ground base monitoring of wind profiler and ceilometer. Thus, those presented observations might be reproducible for other investigation periods of the WAM season.**

45

50

**Additional changes of the authors (AC)**

1. Changes of authors' affiliations:

AC1:
The co-author Karmen Babić has recently changed the affiliation. We added the new one in the manuscript (marked up manuscript p. 1, first line of co-authors).
[a]**now at: Croatia Control Ltd., Velika Gorica, Croatia**

AC2:
The affiliation of co-author Cheikh Dione was not correctly assigned. The affiliation is UPS (number 3) and not KIT (number 2) and he is now working at African Centre of Meteorological Applications for Development. Thus, we changed it accordingly (marked up manuscript p. 1, first line of co-authors).
**Cheikh Dione[3]**
[b]**now at: African Centre of Meteorological Applications for Development, Niamey, Niger**

AC3:
The authors added the current affiliation of Bianca Adler who is now at CIRES and NOAA Physical Sciences Laboratory (see marked up manuscript p. 1, third line of co-authors).
[c]**now at: CIRES, University of Colorado, Boulder, CO, USA**
[d]**now at: NOAA Physical Sciences Laboratory, Boulder, CO, USA**

2. Word substitutions, typos and grammar:

AC4:
Marked up manuscript p. 1, l. 7: The word "**article"** is now used instead of "study"

AC5:
Marked up manuscript p. 1, l. 9-10: The second "low" in the sentence was replaced with the word "marginal".
**Low concentrations of NOx and CO were sampled at Savè supersite near the aircraft measurements and suggested marginal impact of local sources during the case study.**

AC6:
Marked up manuscript p. 1, l. 13-15: There has been a typo so that "to" was removed in l. 14.
**Further, calculated vertical transects of wind speed and BC presume that the observed BC layer was transported from south with maritime inflow, but was**

**mixed vertically after the onset of a nocturnal low-level jet at the measurement site.**

AC7:
Marked up manuscript p. 1, l. 15: "The study" was changed to "**This report** (…)"

AC8:
Marked up manuscript p. 2, l. 15: "The impact of (…)" was changed to "**Investigating the** impact of (…)"

AC9:
Marked up manuscript p. 3, l.17: "The distribution in (…)" was changed to "**Subsequently, BC** distribution within (…)"

AC10:
Marked up manuscript p. 4, l. 1: "previous studies of ALADINA (…)" was changed to "previous studies **using** ALADINA (…)"

AC11:
The sentence was rephrased on p. 4, l. 16-19 (marked up manuscript).
**One reason for this might be damage of optical parts after transportation. But adjusting and calibration was not possible at the research site due to missing laboratory, so that true causes could not be identified.**

AC12:
Marked up manuscript p. 5, l. 3: the typo Element was changed to **element**

AC13:
Marked up manuscript p. 5, l. 5: "**as**" was inserted before specified

AC14:
Marked up manuscript p. 5, l. 22-23: "in comparison with" was changed with "**with respect to**"

AC15:
Marked up manuscript p. 5, l. 24-26: The sentence was rephrased to "**By taking into account that the change in the attenuation ($\sigma$ATN) is caused by an increase of BC mass load on the filter stripe in a corresponding time interval ($\Delta$t), and then BC mass concentration can be derived as follows** (..)

AC16:
Marked up manuscript p. 6, l. 16: "concentration, partially higher than" was replaced with "**concentrations that partially exceed**"

AC17:
Marked up manuscript p. 7, l.13-14: the sentence was newly sorted

AC18:
Marked up manuscript p. 7, l. 31: The word "**Thus**" was inserted in the very beginning of the sentence.

AC19:
Marked up manuscript p. 7, l. 32: The word "**But**" was added in the very beginning of the sentence.

AC20:
Marked up manuscript p. 8, l. 33: "components of the wind" was changed to "**wind components**"

AC21:
Marked up manuscript p. 9, l. 4: "**wind**" was inserted before direction

AC22:
Marked up manuscript p. 9, l. 8: "3.50 m above ground level (a.g.l.)" was reduced to **3.5 m**

AC23:
Marked up manuscript p. 12, l. 10-12: In our opinion, there is a grammar error. We changed the sentence accordingly.
**BC can be separated into fresh BC, aged BC in the Aitken and aged BC in the accumulation mode, allowing to discussing potential sources of BC. Further, simulations allow to embedding the local observations into larger spatial scales.**

AC24:
Marked up manuscript p. 12, l. 24 artifact → artefact

AC25:
Marked up manuscript p. 13, l. 1 artifact → artefact

AC26:
Marked up manuscript p. 16, l. 5-8: The reference of Adon et al. was updated.

Adon, A. J., Liousse, C., Doumbia, E. T., Baeza-Squiban, A., Cachier, H., Léon, J.-F., Yoboué, V., Akpo, A. B., Galy-Lacaux, C., Guinot, B., Zouiten, C., Xu, H., Gardrat, E., and Keita, S.: Physico-chemical characterization of urban aerosols from specific combustion sources in West Africa at Abidjan in Côte d'Ivoire and Cotonou in Benin in the frame of the DACCIWA program, Atmos. Chem. Phys., 20, 5327–5354, https://doi.org/10.5194/acp-20-5327-2020, 2020

AC27:

10 Marked up manuscript p. 16, l. 15-22: The references of Babic et al. (2019a,b) were sorted in the correct order.

[revised manuscript text omitted]